# Benchmarking Tracking Autopilots for Quadrotor Aerial Robotic System Using Heuristic Nonlinear Controllers

**Muhammad Bakr Abdelghany** [1,2,*] **, Ahmed M. Moustafa** [2] **and Mohammed Moness** [2]

1 Group for Research on Automatic Control Engineering, Department of Engineering, University of Sannio, Piazza Roma 21, 82100 Benevento, Italy
2 Computer and Systems Engineering Department, Faculty of Engineering, Minia University, Minia 61111, Egypt
* Correspondence: bmuhammad@unisannio.it

**Abstract:** This paper investigates and benchmarks quadrotor navigation and hold autopilots' global control performance using heuristic optimization algorithms. The compared methods offer advantages in terms of computational effectiveness and efficiency to tune the optimum controller gains for highly nonlinear systems. A nonlinear dynamical model of the quadrotor using the Newton–Euler equations is modeled and validated. Using a modified particle swarm optimization (MPSO) and genetic algorithm (GA) from the heuristic paradigm, an offline optimization problem is formulated and solved for three different controllers: a proportional–derivative (PD) controller, a nonlinear sliding-mode controller (SMC), and a nonlinear backstepping controller (BSC). It is evident through the simulation case studies that the utilization of heuristic optimization techniques for nonlinear controllers considerably enhances the quadrotor system response. The performance of the conventional PD controller, SMC, and BSC is compared with heuristic approaches in terms of stability and influence of internal and external disturbance, and system response using the MATLAB/SIMULINK environment. The simulation results confirm the reliability of the proposed tuned GA and MPSO controllers. The PD controller gives the best performance when the quadrotor system operates at the equilibrium point, while SMC and BSC approaches give the best performance when the system does an aggressive maneuver outside the hovering condition. The overall final results show that the GA-tuned controllers can serve as a benchmark for comparing the global performance of aerial robotic control loops.

**Keywords:** heuristic algorithms; modeling; nonlinear sliding mode control; nonlinear backstepping control; PD; quadrotor

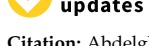



## 1. Introduction

The quadrotor UAVs have gained considerable attraction from the control engineering community in the last decade. Compared with traditional vertical take-off and landing robots, they are more stable and can perform tasks with complete autonomy, minimizing the need for a direct human operator control [1]. They also provide an excellent platform for research and commercial applications, with tremendous technological capabilities in microfeature transducers, actuators, and high-performance computers depending mainly on the mini-, micro-, and nanoembedded systems [2]. The major applications include aerial media video streaming for coverage [3], power line inspections [4], traffic management system [5], spraying and monitoring of crops [6], surveillance [7], and fire detection system and control [8].

Unlike traditional UAVs, the quadrotor consists of two rods along with four actuators. Amongst the four rotors, its two pairs are rotating in a clockwise manner, while the other two are in a counterclockwise direction to balance the torque [9,10]. Hovering and vertical take-off and landing (VTOL) capabilities are prerequisites to reach the requirements for the

applications mentioned above [11]. Traditional UAVs with hovering and VTOL capabilities have a complex structure that is more costly and very difficult to be controlled [10]. As a multivariable nonlinear six-degree-of-freedom (DOF) system, the quadrotor is considered an exciting benchmark in control engineering for the testing, validation, and authentication of control laws tasks in real-time platforms. The quadrotor serves as a helicopter with some significant simplifications, as it is characterized by being underactuated, it has a high-coupling nature, is highly nonlinear, and has strong system uncertainties [12]. The design of autonomous, robust, and computationally effective controllers for such systems is still an open and challenging task for the control engineering community.

### 1.1. Motivation and Literature Review

Several research groups have studied and investigated original approaches for the guidance, navigation, and control of quadrotors. In [12–14], an omnidirectional stationary flying outstretched robot (OS4) quadrotor model was used. It was originally developed at the Ecole Polytechnique Federal De Lausanne (EPFL) to test and validate the proposed control laws to prove their novelty. The X4 Flyer Mark *II* model was an inventive design using inverted teetering rotors developed by The Australian National University [15–18], while Mesicopter (developed by off-the-shelf components) was the invention of Stanford University [19]. STARMAC (*I* and *II*), the Stanford autonomous rotorcraft prototype for multiagent control, was developed to investigate the aerodynamic influences on the quadrotor system when it worked outside the hovering region [10,20]. The PIXHAWK (ETH Zurich) work is commendable and novel for a microquadrotor in terms of its hardware, system software, and building an onboard computer vision system [21]. Table 1 illustrates the popular quadcopter platforms developed and investigated by universities and companies.

**Table 1.** Quadrotor platforms.

| No. | Reference | Platform | University/Company | Year |
|---|---|---|---|---|
| 1 | [21] | PIXHAWK | ETHZ | 2011 |
| 2 | [19] | Mesicopter | Stanford | 2001 |
| 3 | [12] | OS4 | EPFL | 2004 |
| 4 | [20] | STARMAC | Stanford | 2005 |
| 5 | [22] | RAVEN | MIT | 2008 |
| 6 | [15] | X4 Flyer | ANU | 2008 |
| 7 | [23] | MAVs | TUM | 2012 |
| 8 | [24] | GRASP | GRASP team | 2012 |
| 9 | [25] | Parrot AR.DRONE 2.0 | French company Parrot | 2016 |
| 10 | [26] | mdMAPPER1000 | German company MICRODRONES | 2017 |
| 11 | [27] | FlyBebop | Poznan University of Technology | 2020 |

Several studies have investigated the stability and control performance of quadcopter systems using the linear PD control method. For instance, in [13,28], a control strategy for quadrotor-type aerial robots based on the PD loop was developed to stabilize and control the position of a quadrotor during disturbances. Moreover, in [17], the PD controller was used to stabilize the Newton–Euler angles along with the altitude of the quadcopter system. The implementation of the PD controller on a rotorcraft unmanned microaerial vehicle to control the rotation angles and the attitude stability of the quadrotor system on a flying platform was described in [29]. The authors in [30] explained the dynamical model of a quadrotor and implemented the PD control algorithm to regulate the orientation and trajectory tracking of the quadrotor at slow speeds. Moreover, in [31], an extended Kalman filter (EKF) based smart self-tuning fuzzy-PID (SSTF-PID) controller for posture control of the quadcopter was presented. In [32], the authors implemented an adaptive pole placement based on a self-tuning-PID (ST-PID) controller to stabilize the Euler angle of the quadcopter. In [33], a gain-scheduling methodology based on a state feedback control (SFC) in the quadrotor linear model nearby the hovering point was illustrated and the characteristics of

the quadrotor linear model were analyzed. In order to stabilize a quadcopter and to tune the PID parameters, a gain-scheduled PID controller and a fuzzy logic control (FLC) were applied in [34]. Moreover, in that work, they compared the behavior of the conventional PID and the gain-scheduled PID controllers. In [35], a gain-scheduled PID controller scheme connected to the quadcopter helicopter plummeting an endorsed load at an assigned time was presented. In [36], an optimal design of a ducted multipropeller configuration for the aerodynamic performance of the quadrotor drone was integrated. In [37], the authors validated their proposed optimal PID controller by stabilizing the attitude and altitude of the quadrotor system. A novel PID control algorithm was proposed to stabilize the altitude and rotation angles of the quadcopter helicopter landing on a ship deck in [38]. In [39], the authors implemented a multiloop control technique to control and stabilize the position control of a quadcopter with the help of a nonlinear backstepping controller (BSC) and a PID controller to stabilize the orientation angles and altitude for the inner and outer loops, respectively. A study comparing various control techniques such as PID and linear–quadratic regulator (LQR) controllers for modeling the dynamics of quadrotors was presented in [40]. Moreover, an implementation and comparison between the PID and the MRAC (model reference adaptive control) to stabilize the Euler angles of the quadcopter helicopter were proposed in [41].

Due to the nonlinear nature of the quadrotor system, nonlinear control algorithms have been investigated for flight controllers. Some nonlinear control algorithms such as sliding-mode control for fault-tolerant control [42], robust LPV control [43], decentralized sliding-mode control [44], and nonlinear model predictive control (MPC) [45] have been widely used. SMC (sliding-mode control) and BSC were implemented in [28] to solve the stabilization and trajectory tracking problems of the quadcopter system. Furthermore, a system design for measuring UAV parameters was also proposed in [46] in order to aid the tuning of flight control algorithms.

In [14], the authors proposed nonlinear SMC and nonlinear BSC methods to model and control the OS4 quadrotor model by including a nonlinear control design NCD from the MATLAB optimization toolbox. In [47], an implementation of SMC to stabilize the quadrotor and a comparison in terms of performance between an integral-SMC and a model-based reinforcement learning (MBRL) were proposed. The flight controller using multiple-loop SMC and the sliding-mode disturbance observer was designed in [48]. The adaptive sliding-mode control of robot manipulators with system failures was studied in [49]. Further, in [50], the proposed sliding-mode control was extended using a neuroadaptive control methodology and validated concerning the control goals of improving robust performance against matched uncertainties. In [51,52], the authors used two passive and active fault-tolerant sliding-mode controllers (PF-SMC and AF-SMC) to design the ASCTec Pelican quadrotor, using regular and cascaded SMC by tuning the controllers with an ecological systems algorithm (ESA) and a bioinspired stochastic search algorithm (BSSA). The authors in [53] proposed a novel integrated path planning and sliding-mode trajectory tracking control framework for a quadrotor system, which included model uncertainties and environmental obstacles. In [54], the adaptive neural-network-based nonsingular fast terminal sliding-mode approach was used for the trajectory tracking control problem of a quadrotor in order to ensure the finite-time convergence of the integrated system. More specifically, significant aspects such as uncertainties and external disturbance issues were included. A controller for multiple-input multiple-output (MIMO) uncertain nonlinear systems, based on intelligent adaptive BSC using a radial basis function neural network (RBFNN) was presented in [55] to stabilize the orientation and trajectory tracking of a quadrotor helicopter. A comprehensive review of control methods and techniques for different UAV structures, including quadrotors, can be found in [56]. In [57], the authors presented an educational platform for simulating the quadrotor system that was implemented by using MathWorks Virtual Reality toolboxes in order to emulate different trajectory tracking control. The controller strategies applied for quadrotor platforms could

be clustered into three categories: linear robust control, nonlinear control, and intelligent control [58]. Table 2 summarizes all the results presented above.

**Table 2.** Comparison of the recent and current studies.

| | | Controller | | |
| Reference | Platform | Approach | Tuning Method | Controller Target |
| --- | --- | --- | --- | --- |
| [59] | PIXHAWK | PID + SMC | EKF | Localization |
| [39] | COBRA | PID + BSC | Gazebo | Navigation |
| [60] | LinkQuad | PID, LQR, PID + LQR | ITAE, LQR loop | Robustness |
| [61] | MAV | PID | Neural network | Disturbance rejection |
| [28] | QR-UAV | SMC and BSC | FLC | Payload dropping |
| [14] | OS4 | SMC and BSC | NCD | Stabilzation |
| [47] | STARMAC | SMC and ISMC | MBRL | Outdoor control |
| [51,52] | standard | PF-SMC and AF-SMC | ESA and BSSA | Stablization and navigation |
| [55] | MIMO-quadrotor | PF-SMC and AF-SMC | ESA and RBFNN | Trajectory tracking missions |
| [62] | simulation | Feedforward-adaptive control theory | Adaptive laws | Attitude control |

Although SMC, BSC, and PID controllers have been proposed in the literature for quadrotors' stability and tracking performance, the tuning of these controllers with the latest optimization techniques for better system responses is still an open challenge in the presence of disturbances and uncertainty. Recently, the authors in [63] modified the PSO search algorithm by proposing a new algorithm called modified-PSO for the twin-rotor system to obtain acceptable performance in terms of system responses. Therefore, according to the authors' knowledge, research works that address the tuning of the controller gains based on heuristic algorithms, i.e., MPSO and GA, for quadrotor stability and tracking problems, are not available in the literature.

### 1.2. Key Contributions

This paper extends the preliminary study presented in [64] by adopting a more sophisticated control of the quadrotor system. In particular, the stabilization and tracking problems of a quadrotor system are addressed and resolved with the implementation of a PD controller, SMC, and BSC based on heuristic algorithms approaches, i.e., MPSO and GA. Furthermore, the performance of the proposed controllers is tested, analyzed, and authenticated on both linear and nonlinear models of the quadrotor. A comparative study between all the proposed and implemented controllers based on heuristic algorithms is also presented to check the stability and trajectory tracking of the quadrotor system. The main contributions of this paper are summarized as follows:

1. A nested loop controller is proposed and implemented through three different control strategies: PD controller, nonlinear sliding-mode controller, nonlinear backstepping controller;
2. Heuristic optimization tuning algorithms for the proposed control strategies are applied and investigated;
3. A comprehensive benchmark is established for the developed approaches over a nonlinear dynamical quadrotor model.

### 1.3. Outline

The rest of the paper is structured as follows. Preliminaries and the modeling of the quadrotor are derived in Section 2. Section 3 describes the control design of the system. The optimization problem of the controllers is investigated in Section 4. Results showing the effectiveness of the proposed controllers are presented in Section 5. Section 6 concludes the paper. The stability analysis is introduced in Appendix A.

## 2. Preliminaries and Notation

This section helps the reader understand the adopted modeling and control of the quadrotor considered in this study. Some definitions and information for the sake of readers' convenience are also derived in the form of notations and preliminaries.

### 2.1. Notation

Boolean signals, used to illustrate the discrete system, are signals whose values are determined to be false (indicated by zero) and true (indicated by one). Lowercase nonbold letters denote scalars; lowercase bold letters represent column vectors; uppercase nonbold letters indicate matrices and the set of the real numbers is $\mathbb{R}$. In the paper, formulations are derived in a discrete time $k$. The continuous time $t$ can be achieved by $t = kT_s$, with $T_s$ being the sampling time.

### 2.2. System Model

In this subsection, the quadrotor model used for the controller design is developed. Figure 1 shows a six-DOF quadrotor equipped with four rotors. The quadrotor is controlled by varying the angular velocities of rotors, which are rotated by electric motors. Input control forces and moments are produced by changing rotors speeds $(\Omega_f, \Omega_l, \Omega_b, \Omega_r)$.

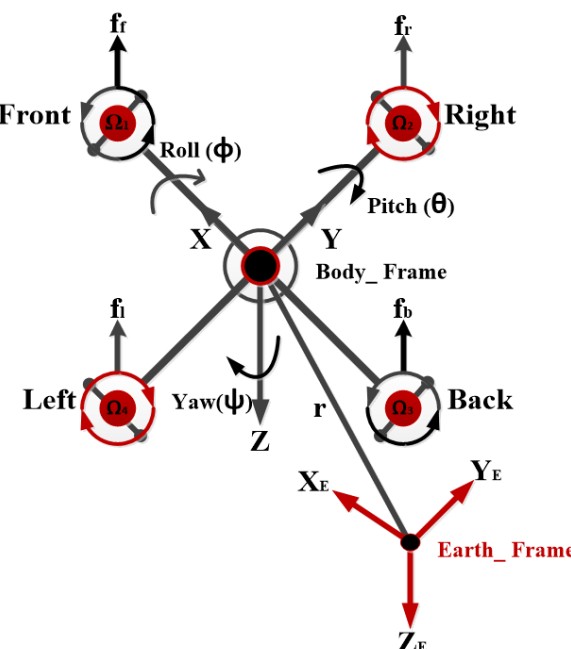

**Figure 1.** A schematic structure of a quadrotor and its relative coordinate systems.

According to the kinematics and dynamics equations, the quadrotor model can be derived based on Newtonian mechanics using the presumptions in [13]. In [65], an approach similar to manipulator identification was applied to estimate the quadrotor model dynamics over an optimized trajectory. The quadrotor model consists of three main parts: motor dynamics, rotor velocity controller, and body dynamics. The quadrotor system open-loop input–output block diagram is presented in Figure 2. The first block corresponding to the motor dynamics consists of four BLDC motors, each with a first-order lag transfer function. The speed rotor to a control input signal conversion is in the second block. The third block consists of two parts: rotational $(\phi, \theta, \psi)$ and translation $(x, y, z)$ subsystems.

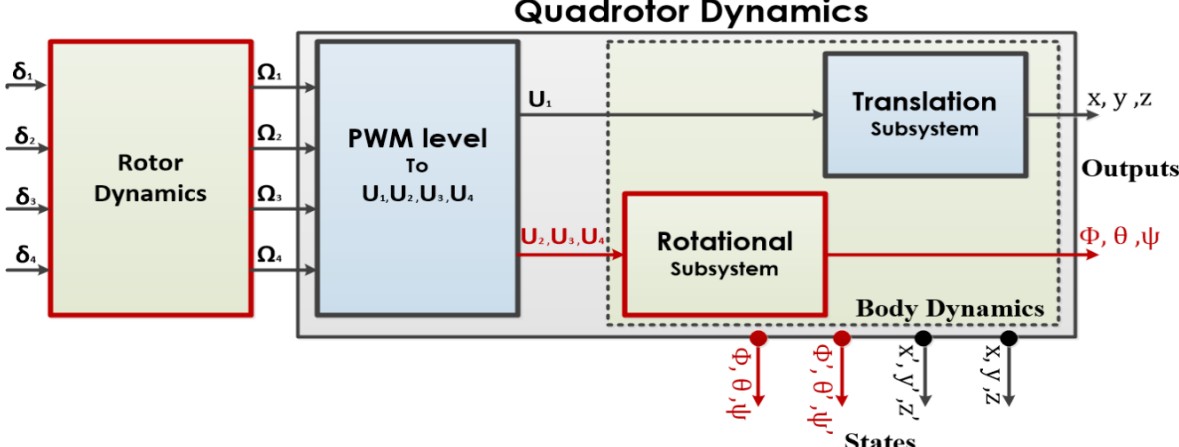

**Figure 2.** Quadrotor system open-loop input–output block diagram.

2.2.1. Kinematics Model

In order to derive the kinematics model, two coordinate frames must be described: the earth inertial frame $(x, y, z)$ and the body-fixed frame $(x_E, y_E, z_E)$ in Figure 1. The two frames are linked by the following rotational matrix

$$R_\Theta = \begin{bmatrix} c\psi c\theta & c\psi s\theta s\phi - s\psi c\phi & c\psi s\theta c\phi + s\psi s\phi \\ s\psi c\theta & s\psi s\theta s\phi + c\psi c\phi & s\psi s\theta c\phi - c\psi s\phi \\ -s\theta & c\theta s\phi & c\theta c\phi \end{bmatrix}, \tag{1}$$

where $R_\Theta$ represents the orientation of the helicopter relative to the earth frame, and $c(.)$ and $s(.)$ denote $\cos(.)$ and $\sin(.)$, respectively [66]. The relation between angular velocities $\omega^{\text{body}} = [p \quad q \quad r]^\top$ in the body frame and the angular velocities in the earth frame ($\dot{\varphi}^{\text{earth}} = [\dot{\phi} \quad \dot{\theta} \quad \dot{\psi}]^\top$) can be defined as

$$\begin{bmatrix} p \\ q \\ r \end{bmatrix} = \begin{bmatrix} 1 & 0 & -s\phi \\ 0 & c\phi & s\phi c\theta \\ 0 & -s\phi & c\phi c\theta \end{bmatrix} \begin{bmatrix} \dot{\phi} \\ \dot{\theta} \\ \dot{\psi} \end{bmatrix}, \tag{2}$$

which can be written into a compact form as $\dot{\varphi}^{\text{earth}} = W_v^{-1} \omega^{\text{body}}$, with $W_v^{-1}$ the directional cosine matrix. Consequently, we can describe the relation between the derivative of the position with respect to the earth frame ($\dot{P} = [\dot{x} \quad \dot{y} \quad \dot{z}]^\top$) and the velocity with respect to the body frame $V \in \mathbb{R}^3$ as $\dot{P} = R_\Theta V$.

2.2.2. Dynamic Model

The dynamic model consists of two parts, a three-degree-of-translation part $(x,y,z)$ and a three-degree-of-rotation part $(\phi,\theta,\psi)$, as illustrated in Figure 2. The translation motion is coupled, while the rotational motion is completely decoupled [13,39,66]. The quadrotor model using the Newton–Euler formulation is given by

$$\begin{bmatrix} F^{\text{body}} \\ \tau^{\text{body}} \end{bmatrix} = \begin{bmatrix} m_q I_{3\times 3} & 0 \\ 0 & I \end{bmatrix} \begin{bmatrix} \dot{v}^{\text{body}} \\ \dot{\omega}^{\text{body}} \end{bmatrix} + \begin{bmatrix} \omega^{\text{body}} \times m_q v^{\text{body}} \\ \omega^{\text{body}} \times \omega^{\text{body}} J_r \end{bmatrix}. \tag{3}$$

The Newton–Euler formulation contains the rotational and translational dynamics of motion. The rotational motion of the quadrotor can be defined using the Newton–Euler method as follows

$$\tau^{\text{body}} = J\dot{\omega}^{\text{body}} + \omega^{\text{body}} \times J\omega^{\text{body}} + [0 \quad 0 \quad J_r\Omega_r]^\top + M_G, \tag{4}$$

where $\tau^{\mathrm{body}} \in \mathbb{R}^{3 \times 1}$ is the total torque, $J \in \mathbb{R}^{3 \times 3}$ is the diagonal inertia matrix, $\omega^{\mathrm{body}}$ are the angular body rates, and $M_G$ are the gyroscopic moments due to the rotors' inertia $J_r$. The rotor's relative speed is given by $\Omega_r = -\Omega_1 + \Omega_2 - \Omega_3 + \Omega_4$. For the sake of simplification, a small angle approximation is made around the hover point where $c(.) = 1$ and $s(.) = 0$ [67,68]. Substituting Equation (1) into (4), the rotational dynamic model can be written as

$$\begin{bmatrix} \tau_\phi \\ \tau_\theta \\ \tau_\psi \end{bmatrix} = \begin{bmatrix} I_x & 0 & 0 \\ 0 & I_y & 0 \\ 0 & 0 & I_z \end{bmatrix} \begin{bmatrix} \ddot{\phi} \\ \ddot{\theta} \\ \ddot{\psi} \end{bmatrix} + \begin{bmatrix} p \\ q \\ r \end{bmatrix} \times \begin{bmatrix} I_x & 0 & 0 \\ 0 & I_y & 0 \\ 0 & 0 & I_z \end{bmatrix} \begin{bmatrix} p \\ q \\ r \end{bmatrix} + \begin{bmatrix} 0 \\ 0 \\ J_r \Omega_r \end{bmatrix}, \tag{5}$$

where $I_x$, $I_y$, and $I_z$ are the inertia moments about the main axes in the body frame. The aerodynamic moments for the $i$th rotor can be described by $\tau_i = c_i(\Omega_i^2)$. The quadrotor's aerodynamic moments with respect to the body frame are

$$\begin{aligned} [\tau_\phi \quad \tau_\theta \quad \tau_\psi]^\top = \quad & [l C_T(-\Omega_2^2 + \Omega_4^2) \quad l C_T(\Omega_1^2 - \Omega_3^2) \\ & C_D(\Omega_1^2 - \Omega_2^2 + \Omega_3^2 - \Omega_4^2)]^\top, \end{aligned} \tag{6}$$

where $C_T$ and $C_D$ denote the thrust and drag factors, respectively, and $\Omega_i$ is the angular velocity of the rotor $i$. Substituting Equation (6) into (5), the rotational subsystem is described by

$$\begin{bmatrix} \ddot{\phi} \\ \ddot{\theta} \\ \ddot{\psi} \end{bmatrix} = \begin{bmatrix} \frac{\dot{\theta}\dot{\psi}(I_y - I_z)}{I_x} - \frac{\dot{\theta}\Omega_r J_r}{I_x} \\ \frac{\dot{\phi}\dot{\psi}(I_z - I_x)}{I_y} + \frac{\dot{\phi}\Omega_r J_r}{I_y} \\ \frac{\dot{\phi}\dot{\theta}(I_x - I_y)}{I_z} \end{bmatrix} + \begin{bmatrix} \frac{U_2 l}{I_x} \\ \frac{U_3 l}{I_y} \\ \frac{U_4}{I_z} \end{bmatrix}, \tag{7}$$

where

$$\begin{aligned} U_1 &= C_T(\Omega_2^2 + \Omega_3^2 + \Omega_4^2 + \Omega_1^2), \\ U_2 &= C_T(-\Omega_2^2 + \Omega_4^2), \\ U_3 &= C_T(\Omega_1^2 - \Omega_3^2), \\ U_4 &= c_D(\Omega_1^2 - \Omega_2^2 + \Omega_3^2 - \Omega_4^2). \end{aligned} \tag{8}$$

By using Newton's second law [67], the translation motion is derived in the earth frame as $m_q \ddot{P} = [0 \; 0 \; m_q \; g]^\top + R_\Theta F^{\mathrm{body}}$, where $P$, $m_q$, and $g$ are the quadrotor's positions, the quadrotor's mass, and the gravitational acceleration, respectively. From [64], the total forces acting on the quadrotor system in the body frame are given by $F^{\mathrm{body}} = [0 \; 0 \; -U_1]^\top$. Indeed, the translation subsystem is described by

$$\begin{bmatrix} \ddot{x} \\ \ddot{y} \\ \ddot{z} \end{bmatrix} = R_\Theta \begin{bmatrix} 0 \\ 0 \\ \frac{-U_1}{m_q} \end{bmatrix} + \begin{bmatrix} 0 \\ 0 \\ g \end{bmatrix} = \begin{bmatrix} \frac{-(c\psi s\theta c\phi + s\psi s\phi)U_1}{m_q} \\ \frac{-(s\psi s\theta c\psi - c\psi s\phi)U_1}{m_q} \\ \frac{g - (c\theta c\phi)U_1}{m_q} \end{bmatrix}^\top. \tag{9}$$

### 2.2.3. Actuation Model

BLDC motors were selected in our quadcopter due to their capability to provide little friction and high torque. A MATLAB identification toolbox was used to identify the model of the BLDC motor. We provided a PWM input signal to each BLDC motor ranging from 25 to 65, which in rpm represents 1000 rpm to 7000 rpm. The identified first-order transfer function for each motor was given in the form of Equation (10), and the relation between angular velocity $\Omega_i$ of each motor and its control input $U_i$ was given by

$$G(s) = \frac{499.7}{s + 4.416}. \tag{10}$$

In order to show the goodness of fit of the BLDC, the fit percent value, which is an accuracy measure that presents how well the model fits the measured data, was defined as

$$\text{Fit}_\% = 100 \times \left( \frac{\|y_{\text{mea}} - y_{\text{sim}}\|}{\|y_{\text{mea}} - \frac{1}{N} \sum y_{\text{mea}}\|} \right) \tag{11}$$

and from Figure 3 it follows that the measured signal fitted the reference signal with a goodness of fit of 80.5%.

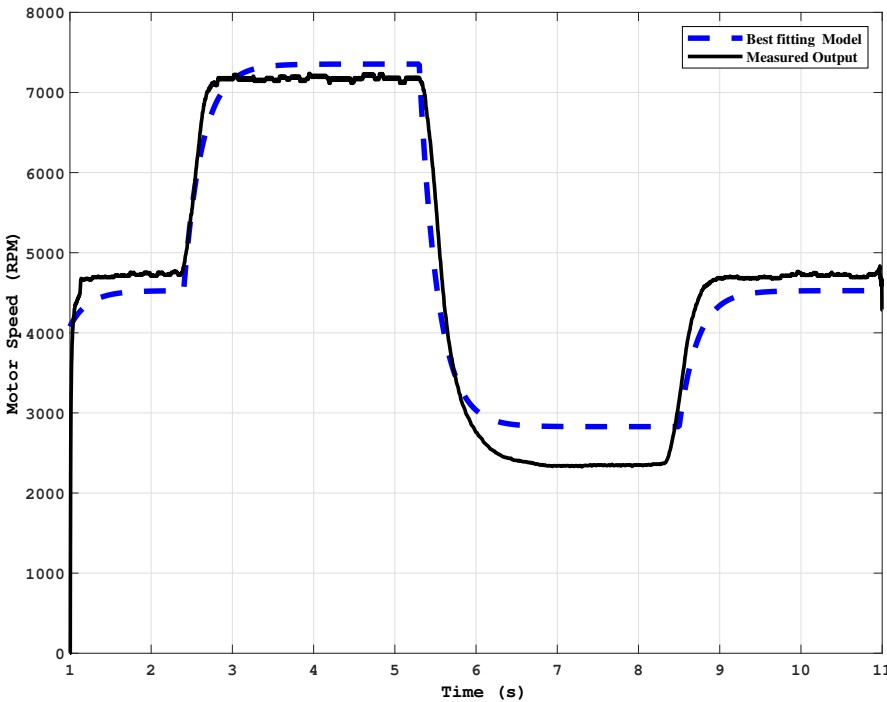

**Figure 3.** Experimental identification of BLDC model.

2.2.4. State Space Model

From Equations (7) and (9), the state space representation for the quadrotor system can be written as follows

$$X = [x_1 \quad x_2 \dots x_{12}]^\top \tag{12}$$

that can be mapped to the DOF of the quadrotor as follows

$$X = [x \quad y \quad z \quad \phi \quad \theta \quad \psi \quad \dot{x} \quad \dot{y} \quad \dot{z} \quad \dot{\phi} \quad \dot{\theta} \quad \dot{\psi}]^\top. \tag{13}$$

The state vector defines the quadrotor states: positions, Euler angles, and linear and angular velocities. The control input vector $U$ is defined as

$$U = [U_1 \quad U_2 \quad U_3 \quad U_4]^\top. \tag{14}$$

From (7) and (9), the nonlinear state space representation can be derived as

$$F(X,U) = \begin{bmatrix} x_1 \\ x_2 \\ x_3 \\ x_4 \\ x_5 \\ x_6 \\ -\frac{(c\psi s\theta c\phi + s\psi s\phi)U_1}{m_q} \\ -\frac{(s\psi s\theta c\psi - c\psi s\phi)U_1}{m_q} \\ \frac{g - (c\theta c\phi)U_1}{m_q} \\ \frac{\dot{\theta}\dot{\psi}(I_y - I_z)}{I_x} - \frac{U_2 l}{I_x} \\ \frac{\dot{\phi}\dot{\psi}(I_z - I_x)}{I_y} + \frac{U_3 l}{I_y} \\ \frac{\dot{\phi}\dot{\theta}(I_x - I_y)}{I_z} + \frac{U_4}{I_z} \end{bmatrix}. \tag{15}$$

## 3. Control Design

In this section, the trajectory tracking problem of the quadrotor system is solved by using three different approaches: a PD controller, SMC, and BSC. The scheme of the control architecture for the quadrotor system is shown in Figure 4. The quadrotor system is a MIMO system with four inputs and six outputs. The reference input vector $R$, the error vector $E$, the control signal vector $U$, and the measured outputs vector $Y$ are given by

$$R = \begin{bmatrix} x_d \\ y_d \\ z_d \\ \phi_d \\ \theta_d \\ \psi_d \end{bmatrix}, U = \begin{bmatrix} U_1 \\ U_2 \\ U_3 \\ U_4 \end{bmatrix}, Y = \begin{bmatrix} x_m \\ y_m \\ z_m \\ \phi_m \\ \theta_m \\ \psi_m \end{bmatrix}, \quad E = \begin{bmatrix} E_x \\ E_y \\ E_z \\ E_\phi \\ E_\theta \\ E_\psi \end{bmatrix}. \tag{16}$$

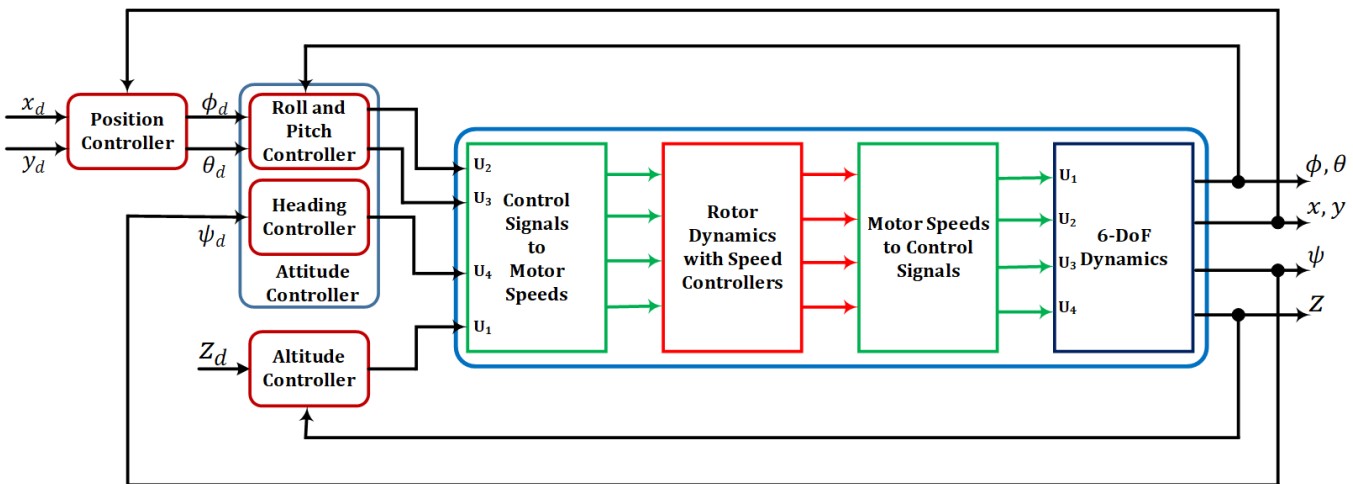

**Figure 4.** Control architecture for the quadrotor platform. The subscript $d$ denotes the desired reference values with respect to the positions and the Euler angles.

On account of the quadrotor dynamics, a nested-loop control methodology is sufficient [13]. From (7) and (9), it is evident that the orientation is independent of the translational motion, but the opposite is not valid. Therefore, an inner-loop control can be developed to guarantee a relaxed trajectory of orientation and altitude, whereas an outer loop handles position control for the quadrotor platform. Three different approaches were used. A PD approach was designed for the outer loop, and all approaches (PD control,

SMC, and BSC) were designed for the inner loop control. For the three control models, a saturation function was required to ensure that the reference control inputs $U_1, U_2, U_3$, and $U_4$ were within the specified limits. The saturation function could be described through the equations

$$\begin{aligned} U_1 &= \text{sat}(U_{1\,\text{max}}), & U_2 &= \text{sat}(U_{2\,\text{max}}), \\ U_3 &= \text{sat}(U_{3\,\text{max}}), & U_4 &= \text{sat}(U_{4\,\text{max}}), \end{aligned} \tag{17}$$

where

$$\text{sat}(x) = \begin{cases} x & \text{if} \quad ||x|| \le x_{\text{max}}, \\ \text{sign}(x)x_{\text{max}} & \text{if} \quad ||x|| > x_{\text{max}}. \end{cases} \tag{18}$$

Figure 4 defines the control scheme of the quadrotor system.

### 3.1. Inner Loop

In order to control the Euler angles and altitude of the proposed quadrotor model, an open-loop simulation was adopted in order to include the attitude and altitude controllers as shown in Figure 4. Furthermore, the attitude and altitude controller took as an input an error signal $e$ which was the difference between the desired values and their actual values. The altitude and the attitude controller produced the output signals $U_1, U2, U3$ and $U4$, respectively. For the sake of clarity, the following subsections present the three controllers, the PD controller, SMC, and BSC controllers, respectively.

### 3.1.1. PD Controller

For the PID approach, the standard PD controller is represented in the frequency domain by

$$C(s) = k_p \left(1 + \frac{1}{T_i s} + \frac{T_d}{\frac{T_d}{N}s + 1}\right) E(s) \tag{19}$$

and the parallel PID controller in the frequency domain is given by

$$C(s) = \left(k_p + \frac{k_i}{s} + k_d s\right) E(s), \tag{20}$$

where $k_i = \frac{k_p}{T_i}$ is the integral gain, $k_d = k_p T_d$ is the derivative gain, and constant $N \ge 10$ for a realizable standard PD controller. Say $U_1, U_2, U_3$, and $U_4$ are the desired control inputs for the quadrotor calculated from the PD approach. Usually, unstable dynamics require $k_i = 0$. Both the standard PD and parallel PD forms were used to stabilize and control the attitude (roll angle $\phi$, the pitch angle $\theta$, the yaw angle $\psi$) and altitude $z$. The control signal $U_1$ was responsible for the altitude $z$ and $U_2, U_3$, and $U_4$ were responsible to stabilize the attitude $(\phi, \theta, \psi)$ as follows

$$U_1 = k_{pz}(z - z_d) + k_{d_z}(\dot{z} - \dot{z}_d), \tag{21a}$$

$$U_2 = k_{p\phi}(\phi - \phi_d) + k_{d\phi}(\dot{\phi} - \dot{\phi}_d), \tag{21b}$$

$$U_3 = k_{p\theta}(\theta - \theta_d) + k_{d\theta}(\dot{\theta} - \dot{\theta}_d), \tag{21c}$$

$$U_4 = k_{p\psi}(\psi - \psi_d) + k_{d\psi}(\dot{\psi} - \dot{\psi}_d), \tag{21d}$$

where $z_d, \phi_d, \theta_d$, and $\psi_d$ are the desired altitude, roll, pitch, and yaw angles, respectively, $\dot{z}_d, \dot{\phi}_d, \dot{\theta}_d$, and $\dot{\psi}_d$ are the desired height, roll, pitch, and yaw angles' rate of change, respectively, and $(k_{pz}, k_{p\phi}, k_{p\theta}, k_{p\psi})$, $(k_{iz}, k_{i\phi}, k_{i\theta}, k_{i\psi})$, and $(k_{dz}, k_{d\phi}, k_{d\theta}, k_{d\psi})$ are proportional gains, integral gains, and derivative gains of altitude and orientation angles control, respectively.

### 3.1.2. Sliding Mode Control

Because of the nonlinearity and coupled dynamics of the quadrotor model, linear control approaches are not sufficient. This section derives an SMC approach, also called the variable structure control (VSC) method. The main idea behind SMC is the possibility

to force the system state trajectory on a chosen sliding surface $S$. The corrective control function targets reaching the sliding surface by handling state trajectory variations. The equivalent control conversely assures that the time derivative of the sliding surface is continuous until it reaches the value of zero, with the aim that the state trajectories remains on the sliding surface [13,28]. The control signal $U(t)$ in the SMC approach consists of two parts that are the equivalent control signal $U_{\text{eq}}$ and corrective control signal $U_{\text{co}}$, as follows

$$U(t) = U_{\text{eq}}(t) + U_{\text{co}}(t). \tag{22}$$

Here, we only present the altitude controller. Hereinafter, the attitude controller can be derived using the same steps. For a fair presentation, the nonlinear system under investigation was extracted from the state space model derived in (15). The SMC for the altitude was designed to converge the altitude $z$ position to its desired value $z_d$. The altitude error signal was given by

$$e_z = z_d - z. \tag{23}$$

The sliding surface was defined as

$$S_z = k_{z_1}(z_d - z) + (\dot{z}_d - \dot{z}), \tag{24}$$

where $k_{z_1} > 0$. The time derivative of the sliding surface yielded

$$\dot{S}_z = k_{z_1}(\dot{z}_d - \dot{z}) + (\ddot{z}_d - \ddot{z}). \tag{25}$$

Moreover, the Lyapunov control theory was used to retain the error on the sliding surface $s(e, t) = 0$. A suitable Lyapunov function was defined by

$$V(S_z, e_z) = 0.5(S_z^2 + e_z^2). \tag{26}$$

By using the guideline in [67], the derivative of the sliding surface was equated to the following exponential reaching law as follows

$$\dot{S}_z = -k_{z_1}\,\text{sign}(S_z) - k_{z_2}, \tag{27}$$

where

$$\text{sign}(S_z) = \begin{cases} 1 & \text{if} \quad S_z > 0 \\ -1 & \text{if} \quad S_z < 0 \end{cases} \tag{28}$$

and $k_{z_1}, k_{z_2} > 0$ are constant numbers. Substituting (27) into (25) yielded

$$k_{z_1}(\dot{z}_d - \dot{z}) + (\ddot{z}_d - \ddot{z}) = -k_{1z}sign(S_z) - k_{2z}S_z. \tag{29}$$

Substituting (15) into (29), the control input $U_1$ could be written as

$$U_1 = \frac{m_q(k_{z_1}\dot{z} + k_{z_2}\,\text{sign}(S_z) + g)}{\cos\phi\cos\theta}. \tag{30}$$

Similarly, the attitude controller $U_2$, $U_3$, and $U_4$ could be derived to stabilize and converge the Euler angles $\phi, \theta, \psi$ to the desired Euler angles $\phi_d, \theta_d, \psi_d$ as follows

$$U_2 = \frac{(k_{\phi_1}\dot{\phi} + k_{\phi_2}\,\text{sign}(S_\phi) - a_1\dot{\psi}\dot{\theta})}{b_1}, \tag{31}$$

$$U_3 = \frac{(k_{\theta_1}\dot{\theta} + k_{\theta_2}\,\text{sign}(S_\phi) - a_3\dot{\psi}\dot{\phi})}{b_2}, \tag{32}$$

$$U_4 = \frac{(k_{\psi_1}\dot{\psi} + k_{\psi_2}\,\text{sign}(S_\phi) - a_5\dot{\theta}\dot{\phi})}{b_3}, \tag{33}$$

where

$$a_1 = \frac{I_y - I_z}{I_x}, \quad a_3 = \frac{I_z - I_x}{I_y}, \quad a_5 = \frac{I_x - I_y}{I_z}, \quad b_1 = \frac{l}{I_x}, \quad b_2 = \frac{l}{I_y}, \quad b_3 = \frac{1}{I_z}. \quad (34)$$

### 3.1.3. Backstepping Controller

Backstepping control, which is based on the state space representation derived in (15), was derived to stabilize and control the orientation and $z$-position of the quadrotor platform. After this, it was considered a recursive design technique that worked by designing intermediate control laws for the state variables [69]. Unlike other control approaches that lead to linearizing the nonlinear models such as the LQR approach, BSC does not eliminate the system's nonlinearities. The design approach of BSC is systematic. Therefore, we introduced the design technique for the altitude. Roll, pitch, and yaw BSC controllers could be derived correspondingly. The two states $(x_3, x_9)$ from (15) were the altitude $z$ and its rate of change $\dot{z}$. By extracting these states from Equation (15), it yielded

$$x_3 = z, \quad x_9 = \dot{z}. \quad (35)$$

By differentiating the above equation and by using (7), it followed that

$$\dot{x}_9 = \ddot{x}_3 = \ddot{z} = \frac{g - U_1(\cos\phi\cos\theta)}{m_q}. \quad (36)$$

The altitude $z$ was in the strict feedback (sort of triangular) form that presents the following Lyapunov function:

$$V_1 = \frac{L_1^2}{2}, \quad (37)$$

where $L_1$ is the error between the desired and measured $x_{3_d}$ position described as follows

$$L_1 = x_{3_d} - x_3 = z_d - z. \quad (38)$$

The derivative of the previous form yielded

$$\dot{V}_1 = L_1\dot{L}_1 = L_1(\dot{z}_d - \dot{z}). \quad (39)$$

In order to achieve this condition, a positive definite bounding function $W_1 L_1$ was chosen for bounding $V_1$ as follows

$$\dot{V}_1 = L_1(\dot{x}_{3d} - x_9) \leq -c_{z_1} L_1^2, \quad (40)$$

where $c_{z_1} > 0$. The virtual control input could be chosen as

$$\dot{x}_{9d} = x_3 + c_1 L_1. \quad (41)$$

Moreover, a new error variable $L_2$ had to be defined

$$L_2 = \dot{L}_1 - c_{z_1} L_1. \quad (42)$$

By modifying Lyapunov function $V_1$ and its time derivative $\dot{V}_1$ in the new coordinate $(L_1, L_2)$, we obtained

$$\dot{V}_1 = L_1\dot{L}_1 = -L_1 L_2 - c_{z_1} L_1^2. \quad (43)$$

The previous Lyapunov function $V_1$ was modified through the function $V_2$ defined as

$$V_2 = V_1 + 0.5L_2^2 = 0.5(L_1^2 + L_2^2). \quad (44)$$

The time derivative of the above equation was given by

$$\dot{V}_2 = \dot{V}_1 + \dot{V}_2 = L_1 \dot{L}_1 + L_2 \dot{L}_2. \tag{45}$$

Substituting the value of $\dot{x}_9$ from (15) into (45), it yielded the inequality

$$\dot{V}_2 = -L_1 L_2 - c_{z_1} L_1^2 + L_2 (\ddot{x}_{3d} - \dot{x}_{3d} - c_{z_1} L_1) \leq -c_{z_1} L_1^2 - c_{z_2} L_2^2. \tag{46}$$

Moreover, the actual control input $U_1$ could be found for the overall altitude control as follows

$$U_1 = \frac{m_q}{\cos \phi \cos \theta}(L_1 + c_{z_1} \dot{L}_1 - c_{z_2} L_2 + g + \ddot{z}_d), \tag{47}$$

where $c_{z_1}, c_{z_2} > 0$. Similarly, the attitude controllers $U_2$, $U_3$ and $U_4$ could be derived to stabilize and converge the Euler angles $\phi, \theta, \psi$ to the desired Euler angles $\phi_d, \theta_d, \psi_d$ as follows

$$U_2 = \frac{1}{b_1}(L_3 + c_{\phi_1} \dot{L}_3 - c_{\phi_2} L_4 - a_1 \dot{\theta} \dot{\psi} + \ddot{\phi}_d), \tag{48}$$

$$U_3 = \frac{1}{b_2}(L_5 + c_{\theta_1} \dot{L}_5 - c_{\theta_2} L_6 - a_3 \dot{\phi} \dot{\psi} + \ddot{\theta}_d), \tag{49}$$

$$U_4 = \frac{1}{b_3}(L_7 + c_{\psi_1} \dot{L}_7 - c_{\psi_2} L_6 - a_5 \dot{\phi} \dot{\theta} + \ddot{\psi}_d), \tag{50}$$

where $b_1, b_2, b_3, a_1, a_3$, and $a_5$ have the same meaning as the terms in (34), with $c_{\phi_1}, c_{\phi_2}, c_{\theta_1}, c_{\theta_2}, c_{\psi_1}, c_{\psi_2} > 0$.

### 3.2. Outer Loop

After stabilizing the altitude and attitude of the quadrotor system using multivariable controllers as discussed above, a complete position controller was developed based on the PD control for the generation of the desired roll angle $\phi_d$ and the pitch angle $\theta_d$. Then, the tracking of the orientation angles was accomplished through the inner control loop. Based on the desired point $(x_d, y_d)$, the desired $(\ddot{x}_d, \ddot{y}_d)$ accelerations were computed by the position PD controllers and were given by

$$\ddot{x}_d = k_{Px}(x_d - x) + k_{Ix} \int_0^t (x_d - x)dt + k_{Dx}(\dot{x}_d - \dot{x}), \tag{51}$$

$$\ddot{y}_d = k_{Py}(y_d - y) + k_{Iy} \int_0^t (y_d - y)dt + k_{Dy}(\dot{y}_d - \dot{y}), \tag{52}$$

where $(x_d, y_d)$ is the desired position, $(\dot{x}_d, \dot{y}_d)$ is the desired velocity, $k_{Px}, k_{Py}, k_{Ix}, k_{Iy}, k_{Dx}, k_{Dy}$ are the PD controller gains. The horizontal positions $(x, y)$ were neither decentralized nor could they be controlled immediately with any one of the control signals $U_i$, $i = 1, \cdots, 4$. However, the positions $(x, y)$ could be controlled by the Euler angles. The reference angles $(\phi_d, \theta_d)$ could be resolved from (53) and (54) as the below subequations:

$$\ddot{x}_d = \frac{-U_1(\cos \phi_d \sin \theta_d \cos \psi + \sin \phi_d \sin \psi)}{m_q}, \tag{53}$$

$$\ddot{y}_d = \frac{-U_1(\cos \phi_d \sin \theta_d \sin \psi - \sin \phi_d \cos \psi)}{m_q}. \tag{54}$$

Based on the small angle values assumption $(\phi, \theta)$, i.e., $s(\phi_d) = \phi_d$, $s(\theta_d) = \theta_d$, and $c(\phi_d) = c(\theta_d) = 1$ about the equilibrium point, we could simplify Equations (51)–(54) as follows

$$\begin{bmatrix} \ddot{x}_d \\ \ddot{y}_d \end{bmatrix} = \frac{U_1}{m_q} \begin{bmatrix} -\sin \psi & -\cos \psi \\ \cos \psi & -\sin \psi \end{bmatrix} \begin{bmatrix} \phi_d \\ \theta_d \end{bmatrix}, \quad \begin{bmatrix} \phi_d \\ \theta_d \end{bmatrix} = \frac{m_q}{U_1} \begin{bmatrix} \ddot{y}_d \cos \psi - \ddot{x}_d \sin \psi \\ -\ddot{y}_d \sin \psi - \ddot{x}_d \cos \psi \end{bmatrix}. \tag{55}$$

In order to show the correct working and capabilities of the controller, for the reader's convenience, the mathematical formulation of the stability analysis for the nonlinear controllers is reported in Appendix A.

## 4. Optimization Problem

In this section, we formally state an optimization problem for the controller tuning. For the different control algorithms, the tuning of the controller gains was performed by two heuristic optimization algorithms: GA and MPSO. These two algorithms have a stochastic programming strategy that simulates the natural evolution process. We focused on a GA and MPSO to tune the parameters of the three benchmarked control approaches.

The objective of the optimization is assessed based on the constraints, which use the critical error as a performance index with these values set as the control gains. Numerous benchmarks utilize the same performance indices, as described in [70]. The most preferred benchmarks for performance indices to evaluate the performance of a controller are ISE (integral squared error) [71], IAE (integral absolute error), ITSE (integral time squared error), and ITAE (integral time absolute error) [72]. In order to balance the performance of the controller with the operational cost, an aggregate of ITAE and the squared control signals $U_i$ were utilized in the following performance index described as

$$J(e, U) = \int_0^\infty \left( t|e_z(t)| + t|e_\phi(t)| + t|e_\theta(t)| + t|e_\psi(t)| + U_1^2 + U_2^2 + U_3^2 + U_4^2 \right) dt. \quad (56)$$

It was clearly observed that the desired solution to the optimization problem was obtained by minimizing the objective function $J(e, U)$ using a GA and MPSO. With regards to physical limitations, the parameters of the controllers must lie in the interval $[0, 100]$. In order to guarantee convergence and reliability, 10 random trials were carried out for both algorithms. The best gains of the GA and MPSO were evaluated using a series of simulation trials in MATLAB before applying the navigational trajectories. All experiments were run on a PC with an Intel Core (TM) i7-1185G7 HQ 3.0 GHz and a RAM capacity of 32 GB. In order to properly scale the different terms in the cost functions, suitable weights were chosen through a series of simulations. For controller tuning, we tuned the inner control loop for both linear and nonlinear controllers (attitude and altitude) using a GA and MPSO. Afterward, we tuned the outer control loop. As a stopping condition, the error tolerance for the performance index was used, as mentioned in Table 3.

**Table 3.** GA and MPSO parameters.

| Algorithm | Parameter | Value |
|-----------|-----------|-------|
| GA | Population size | 50 |
| | Generation | $N_{var}$ * 100 |
| | Elite count | 2.5 |
| | Crossover fraction | 0.8 |
| | Migration fraction | 0.2 |
| | Migration interval | 20 |
| | Function tolerance | $10^{-6}$ |
| MPSO | Swarm size | $N_{var}$ * 10 |
| | Max iteration | $N_{var}$ * 200 |
| | Min fraction neighbors | 0.25 |
| | Initial swarm span | 2000 |
| | Function tolerance | $10^{-6}$ |

The parameters for the GA and MPSO were selected empirically based on a set of initial runs for a linear quadrotor model to investigate the optimality performance of the algorithm parameters following the recommendations in previous work by the authors for a similar twin-rotor problem [63]. Initial runs for a range of values around the one mentioned above with crossover fractions (CF) of $\{0.6, 0.7, 0.8, 0.9\}$ and different population

sizes of $\{80, 120, 160\}$ were conducted. The results illustrated in Figure 5 show that a GA with a CF of 0.8 and a population size (PS) of 120 had an acceptable performance. Based on the obtained results, we came to a selection for the GA and the corresponding values in the PSO. In Table 4, PI denotes the performance index, FEC denotes the function evaluations count, and CT denotes the computation time.

**Table 4.** Test results for different GA configurations.

| Population Size | Crossover Fraction | Best PI | Mean PI | Standard Deviation | Average FEC | Average CT (min) |
|---|---|---|---|---|---|---|
| 80 | 0.6 | 1.216228 | 1.277688 | 0.056627 | 4080 | 7.50 |
| 80 | 0.7 | 1.087693 | 1.191613 | 0.096759 | 4080 | 8.57 |
| 80 | 0.8 | 1.192226 | 1.305917 | 0.068377 | 4080 | 10.66 |
| 80 | 0.9 | 1.157399 | 1.281824 | 0.091529 | 4080 | 8.55 |
| 120 | 0.6 | 1.179078 | 1.312724 | 0.103165 | 6120 | 11.03 |
| 120 | 0.7 | 1.259124 | 1.298816 | 0.037556 | 6120 | 11.22 |
| 120 | 0.8 | 1.195250 | 1.286108 | 0.108280 | 6120 | 10.47 |
| 120 | 0.9 | 1.178229 | 1.254034 | 0.064195 | 6120 | 10.26 |
| 160 | 0.6 | 1.243172 | 1.360126 | 0.074655 | 8160 | 13.70 |
| 160 | 0.7 | 1.173221 | 1.268387 | 0.071414 | 8160 | 13.79 |
| 160 | 0.8 | 1.119076 | 1.449619 | 0.605782 | 8160 | 13.73 |
| 160 | 0.9 | 1.080213 | 1.219144 | 0.090415 | 8160 | 13.75 |

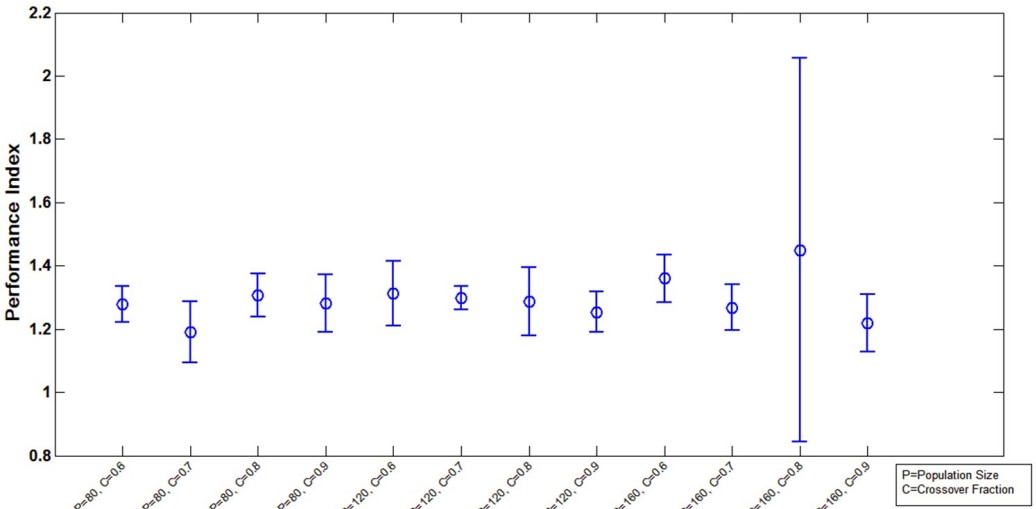

**Figure 5.** Test results for different GA configurations

## 5. Results and Discussion

In this section, the inner and outer loops are designed and investigated for the implemented control approaches: PD controller, BSC, and SMC using the GA and the MPSO optimization algorithms. Table 5 includes the parameters of the quadrotor system utilized in the experiments. In order to authenticate and validate the experiments in this research study, the parameters from the ST-450 experimental quadrotor were adopted.

**Table 5.** Simulation parameters.

| Variable | Description | Value |
|---|---|---|
| $m_q$ | Mass of the quadrotor | 1.006 kg |
| $g$ | Gravitational acceleration | 9.81 ms$^{-2}$ |
| $l$ | Moment arm length | 0.225 m |
| $I_x$ | Moment of inertia along $x$ | 0.0143 kg·m$^2$ |
| $I_y$ | Moment of inertia along $y$ | 0.0148 kg·m$^2$ |
| $I_z$ | Moment of inertia along $z$ | 0.0246 kg·m$^2$ |
| $C_T$ | Thrust factor | $2.2901 \times 10^{-5}$ N·s$^2$ |
| $C_D$ | Drag factor | $1.9318 \times 10^{-7}$ Nm·s$^2$ |
| $J_r$ | Rotor inertia | $6.6231 \times 10^{-5}$ kg·m$^2$ |

*5.1. Optimization Algorithms Performance Analysis*

Each controller for the inner loop was optimized and statistically analyzed over 10 trials for both algorithms to guarantee convergence, stability, and reliability. The step response time-domain characteristics of the best obtained controller with the statistical measures for the optimization processes are reported in Tables 6–9. The bar charts in Figure 6 show the statistical analysis of the optimization results of both algorithms over the 10 trials.

**Table 6.** Optimal tuning results for roll angle control.

| | Best Objective Function | Mean Objective Function | Standard Deviation | Average Computing Time (min) | Rise Time (s) | Overshoot % | Settling Time (s) |
|---|---|---|---|---|---|---|---|
| PD-GA | 3.89187 | 3.90278 | 0.01139 | 10.02 | 0.061 | 0 | 0.111 |
| PD-PSO | 3.89206 | 3.89762 | 0.00723 | 1.93 | 0.061 | 0 | 0.111 |
| BSC-GA | 2.55844 | 2.5618 | 0.00324 | 7.56 | 0.051 | 0 | 0.094 |
| BSC-PSO | 2.56584 | 2.56717 | 0.00134 | 3.7 | 0.052 | 0 | 0.094 |
| SM-GA | 10.13309 | 11.57198 | 1.87204 | 11.21 | 0.138 | 0 | 0.241 |
| SM-PSO | 10.22139 | 11.14991 | 1.06096 | 7.67 | 0.138 | 0 | 0.241 |

**Table 7.** Optimal tuning results for pitch angle control.

| | Best Objective Function | Mean Objective Function | Standard Deviation | Average Computing Time (min) | Rise Time (s) | Overshoot % | Settling Time (s) |
|---|---|---|---|---|---|---|---|
| PD-GA | 3.89187 | 3.90278 | 0.01139 | 19.34 | 0.061 | 0 | 0.111 |
| PD-PSO | 3.89206 | 3.89762 | 0.00723 | 3.89 | 0.061 | 0 | 0.111 |
| BSC-GA | 2.55844 | 2.5618 | 0.00324 | 15.89 | 0.051 | 0 | 0.094 |
| BSC-PSO | 2.56584 | 2.56717 | 0.00134 | 3.99 | 0.052 | 0 | 0.094 |
| SM-GA | 10.13309 | 11.57198 | 1.87204 | 11.22 | 0.138 | 0 | 0.241 |
| SM-PSO | 10.22139 | 11.14991 | 1.06096 | 7.69 | 0.138 | 0 | 0.241 |

**Table 8.** Optimal tuning results for yaw angle control.

| | Best Objective Function | Mean Objective Function | Standard Deviation | Average Computing Time (min) | Rise Time (s) | Overshoot % | Settling Time (s) |
|---|---|---|---|---|---|---|---|
| PD-GA | 39.55843 | 41.58047 | 0.85551 | 8.91 | 0.189 | 0 | 0.337 |
| PD-PSO | 39.43105 | 39.79369 | 1.00944 | 4.66 | 0.189 | 0 | 0.326 |
| BSC-GA | 39.569 | 39.57475 | 0.00631 | 6.45 | 0.198 | 0 | 0.355 |
| BSC-PSO | 39.57773 | 39.5853 | 0.00373 | 4.28 | 0.196 | 0 | 0.351 |
| SM-GA | 52.65907 | 55.06883 | 3.28271 | 4.93 | 0.328 | 0.4 | 0.514 |
| SM-PSO | 52.75427 | 53.77765 | 0.74082 | 3.98 | 0.314 | 0.1 | 0.508 |

**Table 9.** Optimal tuning results for altitude control.

| | Best Objective Function | Mean Objective Function | Standard Deviation | Average Computing Time (min) | Rise Time (s) | Overshoot % | Settling Time (s) |
|---|---|---|---|---|---|---|---|
| PD-GA | 916.05901 | 916.05954 | 0.00051 | 6.24 | 0.461 | 5.9 | 1.142 |
| PD-PSO | 916.06168 | 916.0893 | 0.04788 | 1.92 | 0.461 | 5.9 | 1.142 |
| BSC-GA | 550.49708 | 551.42031 | 2.44355 | 4.34 | 0.466 | 6.5 | 1.184 |
| BSC-PSO | 550.49708 | 550.56088 | 0.08678 | 2.44 | 0.466 | 6.5 | 1.184 |
| SM-GA | 793.10311 | 888.21495 | 111.88848 | 6.14 | 0.754 | 0 | 1.464 |
| SM-PSO | 797.4567 | 823.99621 | 19.36555 | 1.85 | 0.756 | 0 | 1.464 |

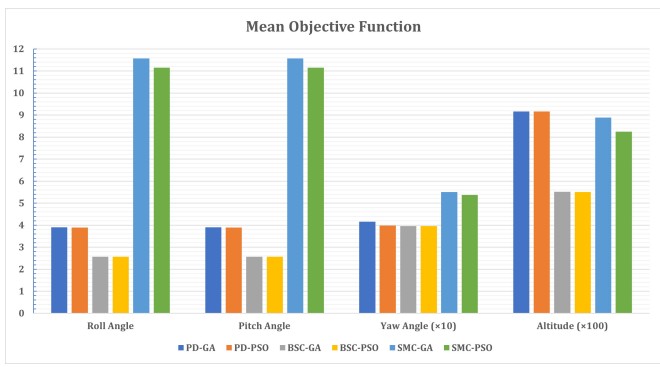
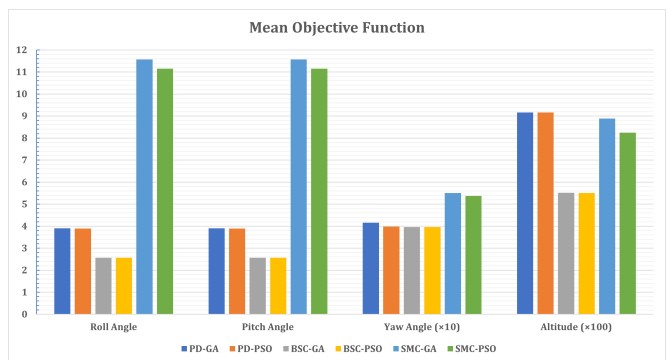
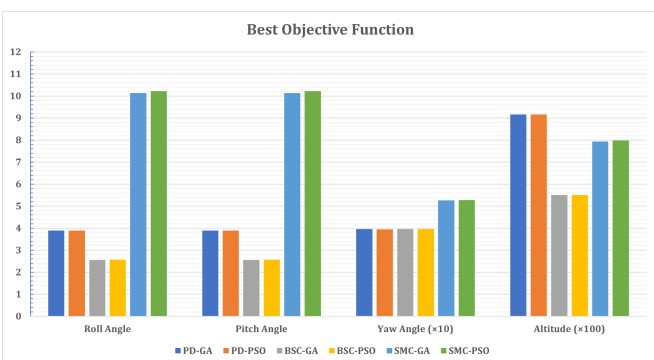
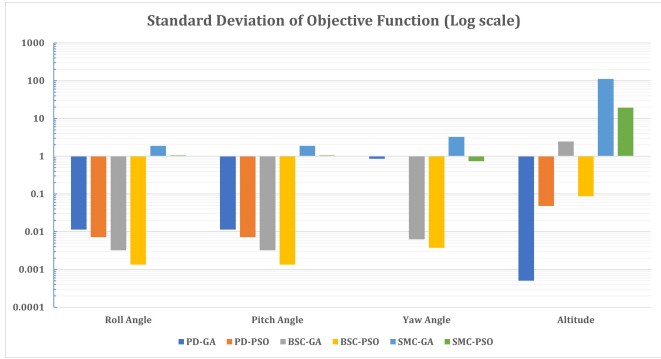

**Figure 6.** Statistical performance bar charts for the optimization of the controllers tuning process.

The best configurations obtained for the three controllers are summarized in Tables 10–12. The step responses are plotted in Figures 7–10 to visually compare the PD controller, the BSC, and the SMC for the inner control loops. The step commands were a 5° step for the orientation angles, and a 2 m command for the altitude. From Figure 8, it can be observed that the BSC-GA and the BSC-PSO controllers gave the best roll angles. Interestingly, the SMC-GA and the SMC-PSO controllers successfully tracked the step command with a slower response time.

**Table 10.** Best obtained PD controller parameters of GA and MPSO algorithms after 10 trials.

| | Attitude Control | | | | Heading Control | | Altitude Control | | Position Control | | | |
|---|---|---|---|---|---|---|---|---|---|---|---|---|
| | $k_{p\phi}$ | $k_{d\phi}$ | $k_{p\theta}$ | $k_{d\theta}$ | $k_{p\psi}$ | $k_{d\psi}$ | $k_{pz}$ | $k_{dz}$ | $k_{px}$ | $k_{dx}$ | $k_{py}$ | $k_{dy}$ |
| PD-GA | 100 | 3.21 | 100 | 3.21 | 7.76 | 0.86 | 100 | 20.84 | 10.12 | 5.97 | 10.61 | 6.07 |
| PD-PSO | 100 | 3.21 | 100 | 3.21 | 6.59 | 0.75 | 100 | 20.84 | 10.06 | 5.92 | 11.98 | 6.60 |

**Table 11.** Best obtained SMC controller parameters of GA and MPSO algorithms after 10 trials.

| | Attitude Control | | | | Heading Control | | Altitude Control | | Position Control | | | |
|---|---|---|---|---|---|---|---|---|---|---|---|---|
| | $k_{\phi_1}$ | $k_{\phi_2}$ | $k_{\theta_1}$ | $k_{2\theta_2}$ | $k_{\psi_1}$ | $k_{\psi_2}$ | $k_{z_1}$ | $k_{z_2}$ | $k_{x_1}$ | $k_{x_2}$ | $k_{y_1}$ | $k_{y_2}$ |
| SMC-GA | 92.59 | 72.72 | 92.59 | 72.72 | 17.44 | 17.16 | 100 | 5.15 | 11 | 6.60 | 11.39 | 6.60 |
| SMC-PSO | 90.26 | 96.37 | 70.26 | 96.37 | 18.11 | 17.04 | 5.15 | 100 | 10.56 | 6.25 | 11.88 | 6.87 |

**Table 12.** Best obtained BSC controller parameters of GA and MPSO algorithms after 10 trials.

| | Attitude Control | | | | Heading Control | | Altitude Control | | Position Control | | | |
|---|---|---|---|---|---|---|---|---|---|---|---|---|
| | $c_{\phi_1}$ | $c_{\phi_2}$ | $c_{\theta_1}$ | $c_{\theta_2}$ | $c_{\psi_1}$ | $c_{\psi_2}$ | $c_{z_1}$ | $c_{z_2}$ | $c_{x_1}$ | $c_{x_2}$ | $c_{y_1}$ | $c_{y_2}$ |
| BSC-GA | 13.98 | 99.29 | 13.98 | 99.29 | 3.77 | 18.22 | 3.18 | 68.02 | 7.43 | 4.57 | 8.91 | 5.08 |
| BSC-PSO | 13.99 | 100 | 13.99 | 100 | 4.09 | 20.79 | 3.17 | 49.06 | 6.80 | 4.35 | 8 | 4.71 |

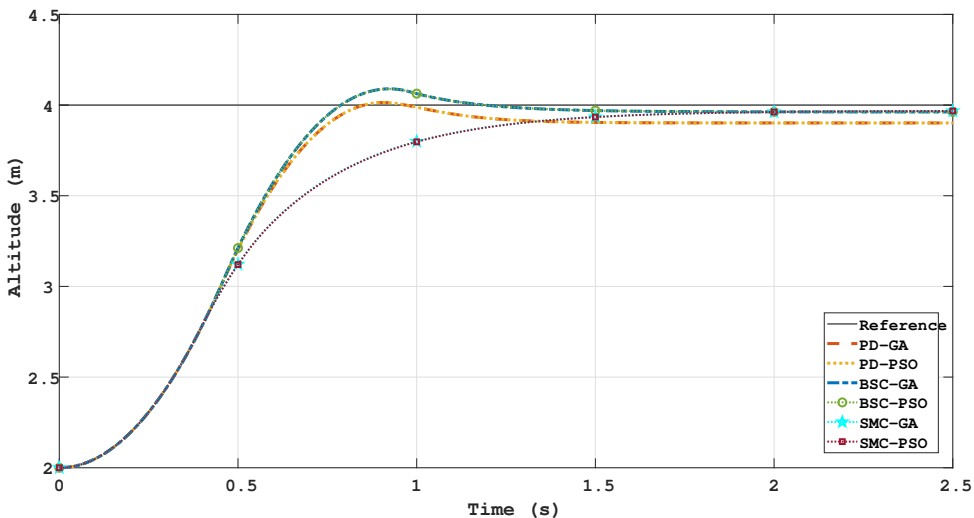

**Figure 7.** Step responses for the altitude controllers.

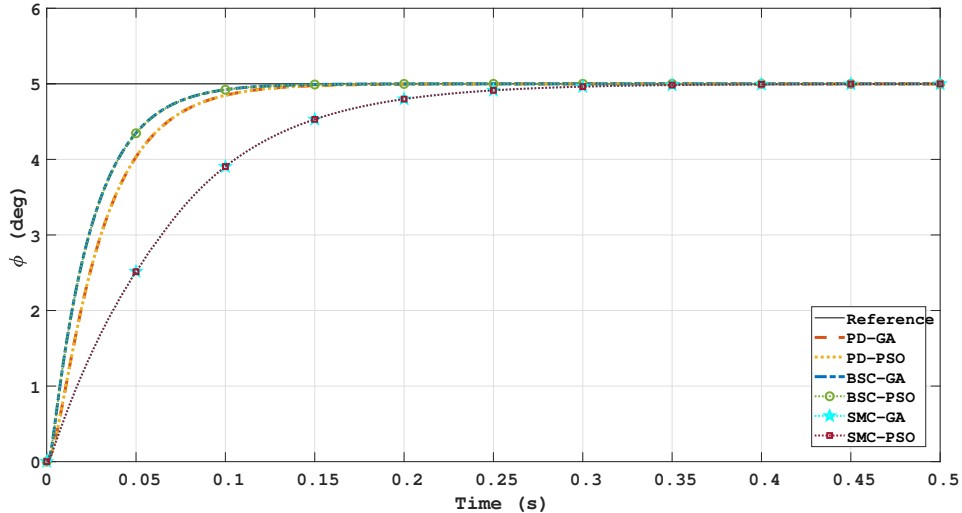

**Figure 8.** Step responses for the roll angle controllers.

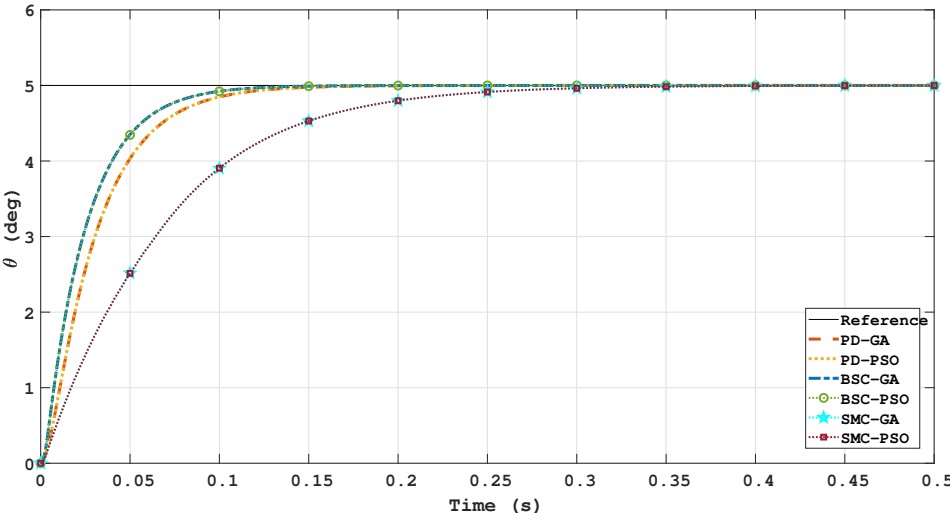

**Figure 9.** Step responses for the pitch angle controllers.

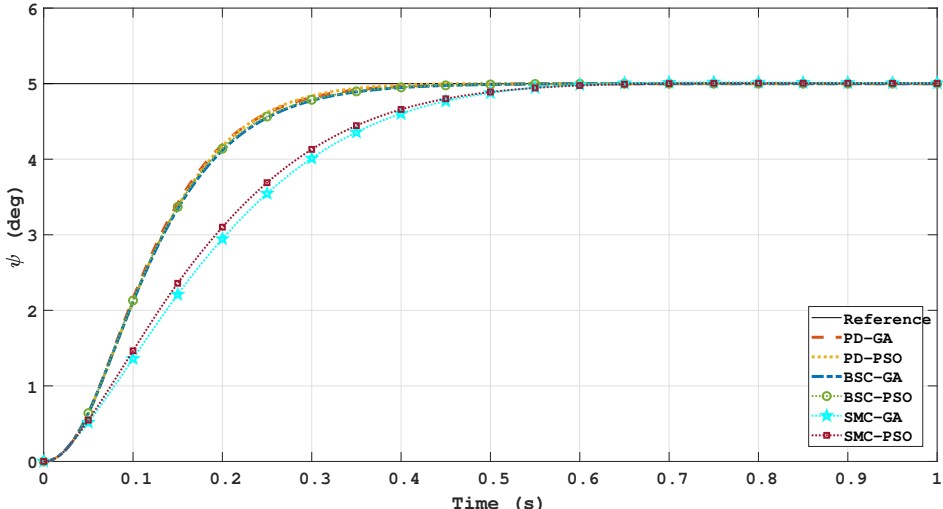

**Figure 10.** Step responses for the yaw angle controllers.

To ensure the stability and tracking performance of the three controllers. The three control configurations were validated against an up–down–up signal. Figures 11–14 illustrate the tracking results of the attitude and altitude. Figures 11–13 show the angles $\{\phi, \theta, \psi\}$ responses based on three different controllers. According to the given input signal, all the implemented controllers were acceptable for tracking the reference signal for the angles $\{\phi, \theta, \psi\}$. On the other hand, from the perspective of the control architecture, it is possible to observe from Figure 14 that the altitude $z$ responses based on the three different controller algorithms correctly met the reference signal for all variable step commands.

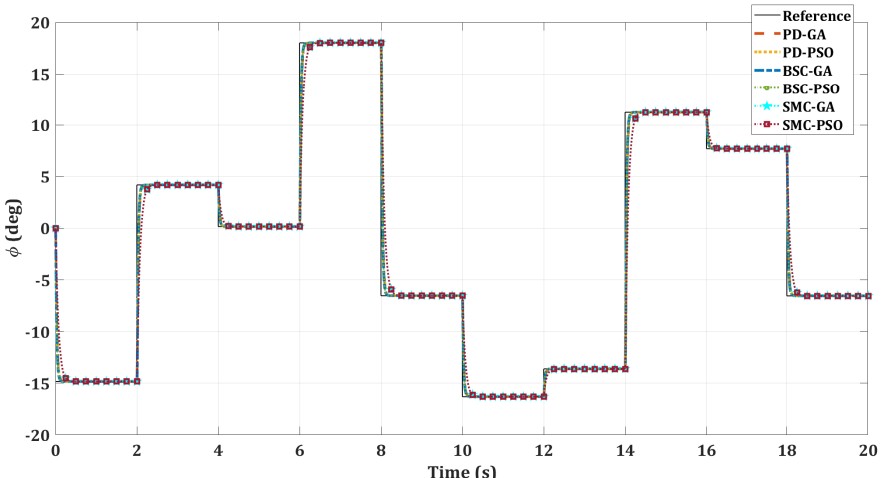

**Figure 11.** The up−down−up response of quadrotor with nonlinear approaches of roll angle.

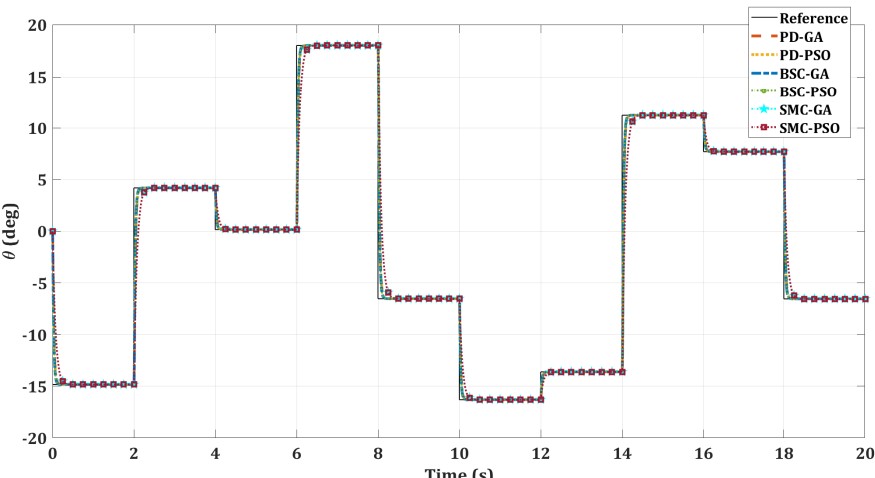

**Figure 12.** The up−down−up response of quadrotor with nonlinear approaches of pitch angle.

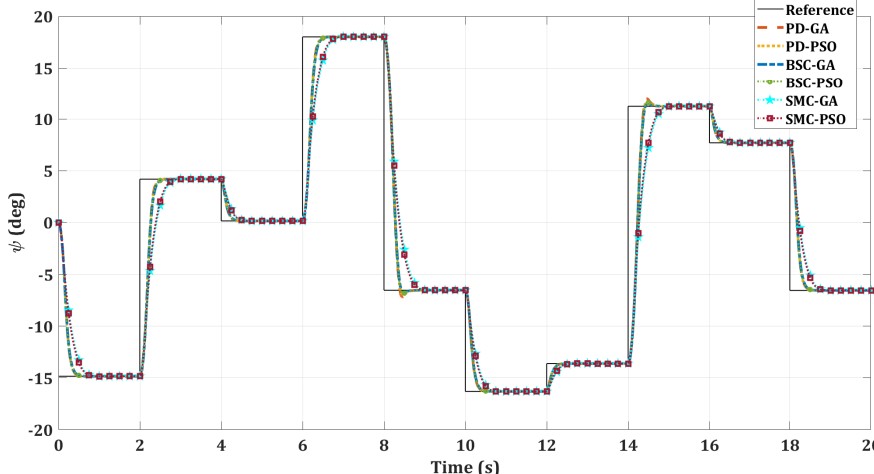

**Figure 13.** The up−down−up response of quadrotor with nonlinear approaches of yaw angle.

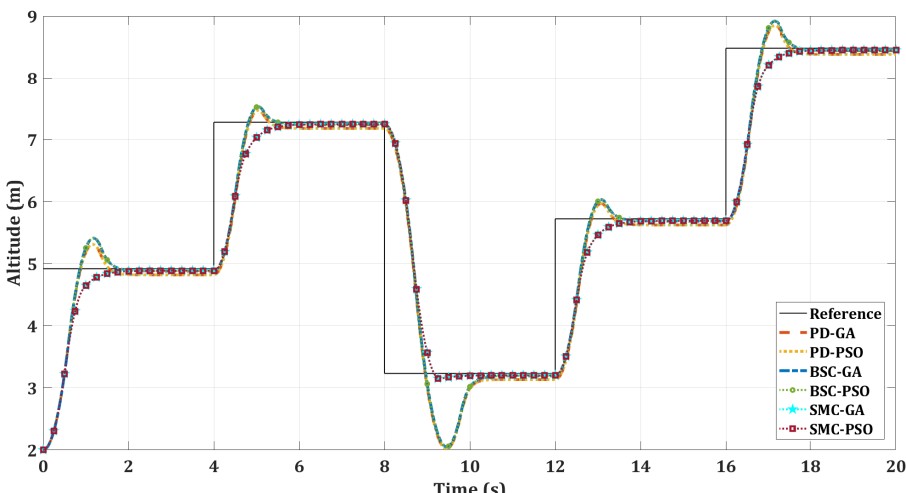

**Figure 14.** The up−down−up response of quadrotor with nonlinear approaches of altitude.

*5.2. Three-Dimensional (3D) Trajectories*

In order to evaluate the proposed control loops, two different 3D trajectories were considered in a windy environment. The wind effect was implemented as noisy forces and moments added to the quadrotor dynamical differential equations of motions. The first trajectory was the curved triangle, which can be mathematically described as

$$\begin{bmatrix} x_d^{\text{tri}}(t) \\ y_d^{\text{tri}}(t) \\ z_d^{\text{tri}}(t) \end{bmatrix} = \begin{bmatrix} A^{\text{tri}} \sin\left(\omega^{\text{tri}} t\right) \\ -A^{\text{tri}} \sin\left(\omega^{\text{tri}} t\right) \cos(\omega t) \\ A^{\text{tri}} \cos\left(\omega^{\text{tri}} t\right) + b^{\text{tri}} \end{bmatrix}, \tag{57}$$

where $A^{\text{tri}} = 6$, $\omega^{\text{tri}} = 0.033$, and $b^{\text{tri}} = -8$. Similarly, the helix trajectory was defined as

$$\begin{bmatrix} x_d^{\text{hex}}(t) \\ y_d^{\text{hex}}(t) \\ z_d^{\text{hex}}(t) \end{bmatrix} = \begin{bmatrix} A^{\text{hex}} \sin\left(\omega^{\text{hex}} t\right) \\ A^{\text{hex}} \cos\left(\omega^{\text{hex}} t\right) \\ c^{\text{hex}} t + b^{\text{hex}} \end{bmatrix}, \tag{58}$$

where $A^{\text{hex}} = 8$, $\omega^{\text{hex}} = 0.1$, $c^{\text{hex}} = -0.2$, and $b^{\text{hex}} = -2$.

For the curved triangle, Figures 15–17 illustrate the tracking performance of the curved triangle trajectory for the PD control, the SMC, and the BSC, respectively. Likewise, Figures 18–20 show the tracking capability for the three controllers following the helix trajectory. It can be noticed that the PD and the SMC followed the trajectories perfectly while the BSC produced a slightly significant tracking error at some waypoints along both trajectories. One of the known disadvantages of BSC is that it generates aggressive control signals. Recalling from the step response, the BSC achieved the fastest response time for the roll angle, but at the expense of more aggressive trajectories. Therefore, the trajectories for the BSC exhibited tracking errors away from reference points at steeper maneuvers.

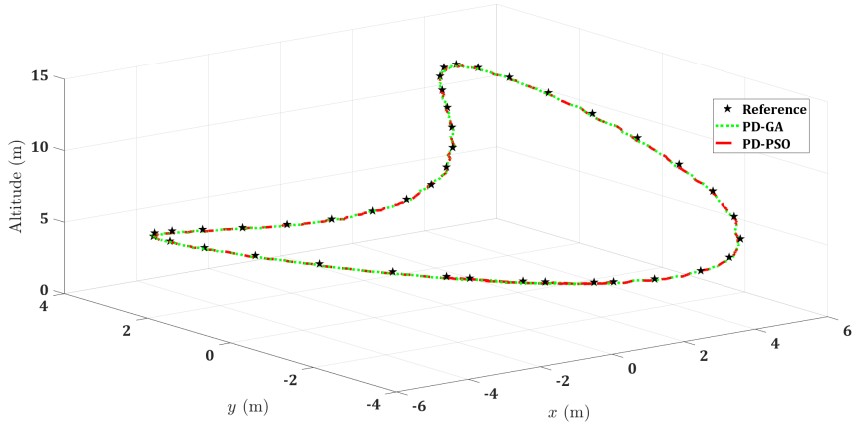

**Figure 15.** PD tracking measures for the curved triangular trajectory

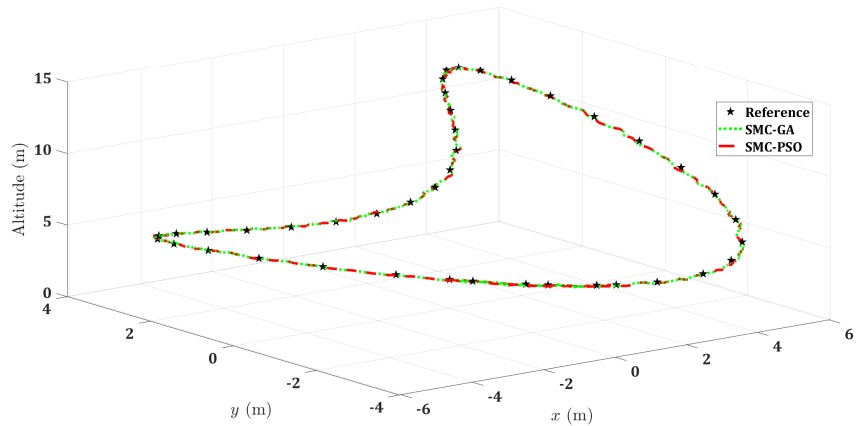

**Figure 16.** SMC tracking measures for the curved triangular trajectory.

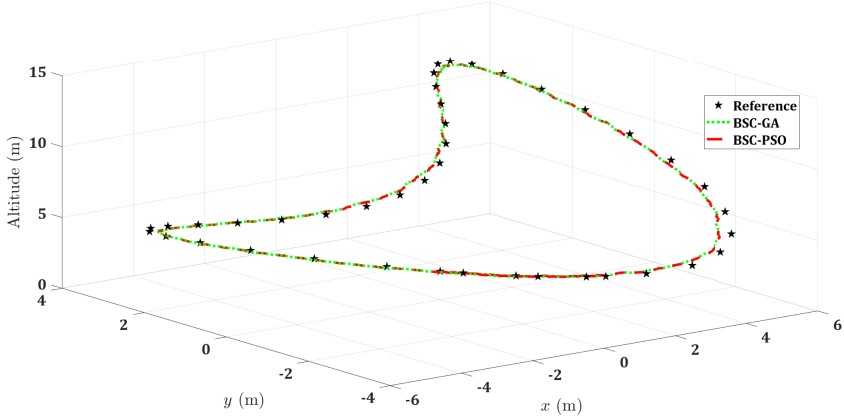

**Figure 17.** BSC tracking measures for the curved triangular trajectory.

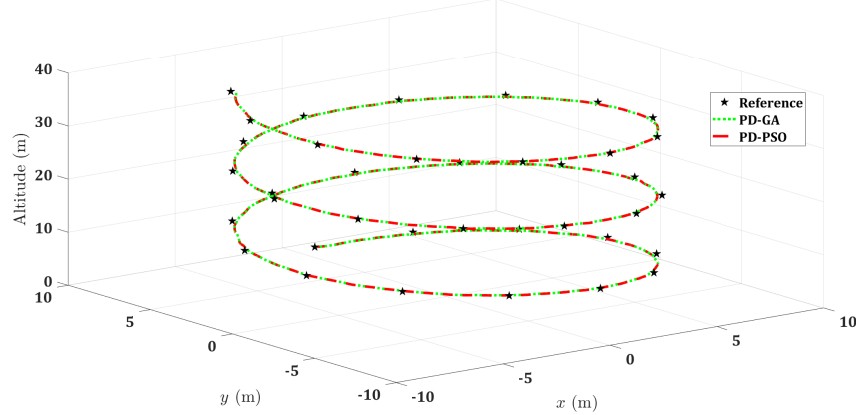

**Figure 18.** PD tracking measures for the helix trajectory.

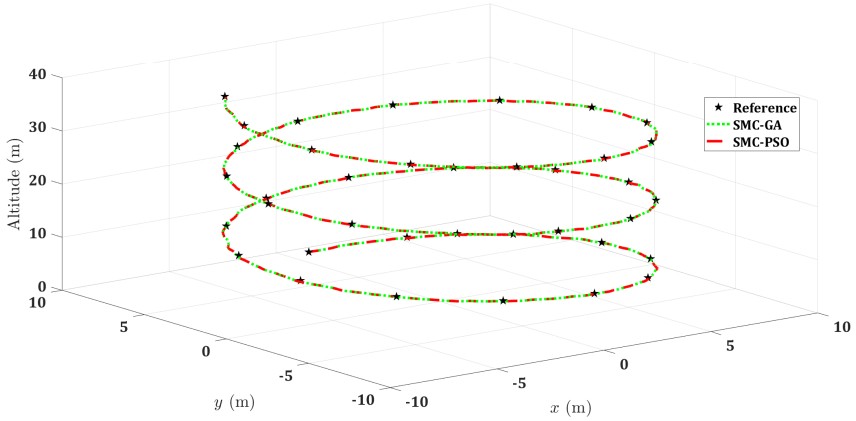

**Figure 19.** SMC tracking measures for the helix trajectory.

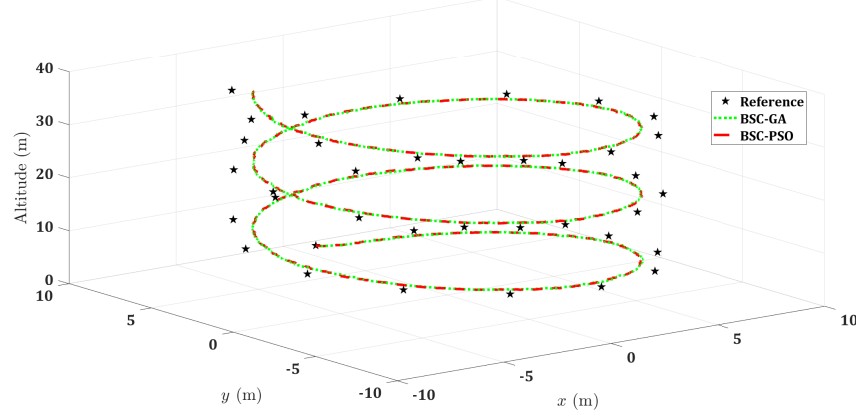

**Figure 20.** BSC tracking measures for the helix trajectory.

For further investigation, the mean square error (MSE), the normalized root-MSE (NRMSE), and the goodness of fit percentage for the tracking performance over the three dimensions were calculated. These statistical measures were defined as

$$\text{MSE} = \frac{1}{N} \sum_{}^{N} (x_d - x)^2, \tag{59}$$

$$\text{NRMSE} = 100 \times \frac{\| x_d - x \|}{\left\| x_d - \frac{1}{N} \sum^N x_d \right\|} \%, \tag{60}$$

$$\text{Goodness} = (100 - \text{NRMSE})\%, \tag{61}$$

respectively. Tables 13 and 14 comprise the computed measures over the $x$, $y$, and $z$ axes for the two trajectories, respectively. The BSC aggressive maneuvering can be noticed on the $y$ dimension for the curved triangle trajectory with an increase of the NRMSE up to 8%. Moreover, the BSC exhibited an NRMSE of 10–11% on the $x$ and the $y$ axes. On the other hand, the PD controller and the SMC showed perfect tracking properties with a goodness of fit measure of 98–99%.

**Table 13.** Statistical tracking measures for the curved triangle trajectory.

| | $x$ | | | $y$ | | | $z$ | | |
| --- | --- | --- | --- | --- | --- | --- | --- | --- | --- |
| | MSE | NRMSE % | Goodness % | MSE | NRMSE % | Goodness % | MSE | NRMSE % | Goodness % |
| PD-GA | 0.0009 | 0.73 | 99.27 | 0.0012 | 1.66 | 98.34 | 0.0097 | 1.13 | 97.72 |
| PD-PSO | 0.0009 | 0.74 | 99.26 | 0.0009 | 1.48 | 98.52 | 0.0097 | 1.13 | 97.72 |
| BSC-GA | 0.0068 | 2.04 | 97.96 | 0.0273 | 8.61 | 91.39 | 0.0014 | 0.43 | 99.12 |
| BSC-PSO | 0.0074 | 2.12 | 97.88 | 0.0246 | 8.14 | 91.86 | 0.0014 | 0.43 | 99.12 |
| SMC-GA | 0.0016 | 0.96 | 99.04 | 0.0011 | 1.63 | 98.37 | 0.0029 | 0.61 | 98.76 |
| SMC-PSO | 0.0018 | 1.02 | 98.98 | 0.0014 | 1.78 | 98.22 | 0.0029 | 0.62 | 98.75 |

**Table 14.** Statistical tracking measures for the helix trajectory.

| | $x$ | | | $y$ | | | $z$ | | |
| --- | --- | --- | --- | --- | --- | --- | --- | --- | --- |
| | MSE | NRMSE % | Goodness % | MSE | NRMSE % | Goodness % | MSE | NRMSE % | Goodness % |
| PD-GA | 0.0188 | 2.45 | 97.55 | 0.0156 | 2.29 | 97.71 | 0.0098 | 0.44 | 99.05 |
| PD-PSO | 0.0191 | 2.47 | 97.53 | 0.012 | 2 | 98 | 0.0098 | 0.44 | 99.05 |
| BSC-GA | 0.3168 | 10.87 | 89.13 | 0.289 | 10.64 | 89.34 | 0.0016 | 0.18 | 99.61 |
| BSC-PSO | 0.3353 | 11.21 | 88.78 | 0.2643 | 10.13 | 89.85 | 0.0016 | 0.18 | 99.61 |
| SMC-GA | 0.0016 | 0.71 | 99.29 | 0.0012 | 0.61 | 99.39 | 0.0089 | 0.42 | 99.09 |
| SMC-PSO | 0.0019 | 0.75 | 99.25 | 0.0014 | 0.67 | 99.33 | 0.0089 | 0.42 | 99.09 |

## 6. Conclusions and Future Prospects

This paper presented three control techniques (PD control, SMC, BSC) that were tuned based on heuristic algorithm approaches (GA and MPSO) to stabilize a nonlinear quadrotor model. Depending on the model states, inner loop control and outer loop control strategies were employed to stabilize and track the orientation and position commands. The MATLAB/SIMULINK environment was adopted to evaluate and compare the implemented controllers with different optimization techniques in their dynamic performances under different inputs and scenarios. The ST-450 experimental quadrotor was used as a benchmark.

A parameter tuning of the implemented controllers was carried out using GA and MPSO, where the performance index guided the cost function. The GA and the MPSO were used, with the cost function count and error tolerance as stopping criteria. The results showed that the performance index within a sufficient search time calculated through the GA optimization was better and contrasted with the MPSO. Furthermore, the statistical examination of 10 independent trials was illustrated to ensure the applicability and reliability of the achieved resolution for these controllers. The PD control performed better than the BSC and the SMC when the quadrotor flew inside the linear region around

the hovering point. Comparatively, the BSC and the SMC showed good performance outside the linear equilibrium range. According to the system's step response tests, the GA showed better results with a good sensible settling time, rise time, and overshoot, than the MPSO. The varying trajectory up–down–up test for the trajectory tracking of the quadrotor was also carried out for a thorough comparison between all three control configurations on the nonlinear model of the quadrotor system. Finally, the 3D trajectories of the curved triangle and helix showed the validity and applicability of the benchmarked controllers. However, the BSC exhibited some aggressiveness in the tracking performance.

Future work includes extending the proposed study to a real-world environment and experiments to get a deeper comparison of their dynamic performance. Additionally, we wish to extend and enhance the system's performance with the design of a computationally effective MPC approach to improve tracking performance and the ability of the quadrotor to carry payloads.

**Author Contributions:** Conceptualization, M.B.A. and A.M.M.; methodology, M.B.A. and A.M.M.; software, M.B.A. and A.M.M.; validation, M.B.A. and A.M.M.; investigation, M.B.A. and A.M.M.; resources, M.M.; writing—original draft preparation, M.B.A.; writing—review and editing, M.B.A., A.M.M. and M.M.; supervision, M.M. All authors have read and agreed to the published version of the manuscript.

**Funding:** This research received no external funding.

**Data Availability Statement:** The data are available upon request from the corresponding author.

**Conflicts of Interest:** The authors declare no conflict of interest.

## Appendix A. Stability Analysis

In this appendix, the state convergence properties of nonlinear controllers are analyzed and are evaluated based on the Lyapunov stability approach. Consider the positive definite Lyapunov function

$$V_i = 0.5 L_i^2, \tag{A1}$$

for $i = 1, 2, \ldots, 8$. The system is analyzed in the transformed coordinate system using nonlinear control for the altitude dynamics as follows

$$\begin{bmatrix} \dot{L}_1 \\ \dot{L}_2 \end{bmatrix} = \begin{bmatrix} c_1 & -1 \\ 1 & c_2 \end{bmatrix} \begin{bmatrix} L_1 \\ L_2 \end{bmatrix}. \tag{A2}$$

The closed-loop system is a nonlinear autonomous system in the coordinates of the error. The stability of such a nonlinear autonomous system can be analyzed based on the Krasovskii–LaSalle formula [73]. Using (44), the Lyapunov function is positive definite, radially unbounded. Using (45), the system is ensured to be stable if the Lie derivative of (A1) is negative semidefinite. Based on Lyapunov's second approach, the system is globally asymptotically stable. Furthermore, the trajectories $L_1$ and $L_2$ are bounded. From (47), the actual control input boundedness can be formulated. Furthermore, the Lyapunov function has a minimum rate of dissipation given by

$$\dot{V}_2(L) \leq -\alpha V_2(L), \tag{A3}$$

where $c_1 \geq 0.5$, $c_2 \geq 0.5$, and $L = \begin{bmatrix} L_1 & L_2 \end{bmatrix}$. Consequently, the system is globally exponentially stable with a bounded control input. Hereinafter, a bound on the error of the transient tracking is derived for the controller gains. Based on (45), the rate of the Lyapunov function satisfies

$$\dot{V}_2(L) \leq -c_1 L_1^2 \tag{A4}$$

and the $L_2$ norm of the transient tracking error is defined as follows

$$
\begin{aligned}
||L_1||^2 &= \int_0^\infty L_1(\tau)^2 d\tau \\
&\leq \frac{-1}{c_1} \int_0^\infty \dot{V}_2(L(\tau)) d\tau \\
&\leq \frac{1}{c_1} [V_2(L(0)) - V_2(L(\infty))] \\
&\leq \frac{1}{c_1} V_2(L(0)).
\end{aligned}
\tag{A5}
$$

Following the same method as the altitude controller, the stability analysis for the attitude nonlinear control approaches is derived.

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
