# Peer review of "Benchmarking Tracking Autopilots for Quadrotor Aerial Robotic System Using Heuristic Nonlinear Controllers"

_drones, doi:10.3390/drones6120379_

Round 1
Reviewer 1 Report (Previous Reviewer 2)
As a re-submitted paper, this paper has been considerably modified, and corrected all problems I mentioned in the previous version. The paper looks fine to me now, and I don't have further comments.
Author Response
We thank the Reviewer for reviewing again the paper and for appreciating it. We are happy that he/she finds the updated version of our paper pleasant to read as it was exactly our initial aim.
Reviewer 2 Report (Previous Reviewer 3)
The paper was significantly corrected and all remarks have been taken into account. I guess the paper may be published in present form.
Author Response
We thank the Reviewer for reviewing again the paper and for appreciating it. We are happy that he/she finds the updated version of our paper pleasant to read. Thanks again for your time and availability despite your busy schedule.

Reviewer 3 Report (New Reviewer)
The authors have presented three control techniques (PD, SMC, BSC) that are tuned based on heuristic algorithm approaches (GA and MPSO) to stabilize a nonlinear quadrotor model. The manuscript is well written with good research gap clarity and a sufficient literature survey. The method and results are also presented in depth and detail with all settings and specifications outlined for other researchers to further investigate this field. The results discussion, findings and conclusion support the work presented and thus I recommend for the editor accept this manuscript for publication.
Author Response
We thank the Reviewer for appreciating the paper. We are happy that he/she accepted its revised version.

Reviewer 4 Report (New Reviewer)
The paper titled: Benchmarking tracking autopilots for quadrotor aerial robotic system using heuristic nonlinear controller
The paper dealt with Heuristic algorithms; modeling; nonlinear sliding mode control; nonlinear backstepping control; PD; quadrotor and so on.
The authors with some numerical simulation results confirmed with success the reliability of the proposed tuned GA and MPSO controllers. The PD controller gives the best performance when the quadrotor system operates at the equilibrium point, while SMC and BSC approaches give the best performance when the system does an aggressive maneuver outside the hovering condition. The overall results show that the GA-tuned controllers can serve as a benchmark for comparing global performance of aerial robotic control loops.
The sections of the proposed paper are well described with relevant results.
The concluding remarks are done according to the obtained results.
The state of the art is ok
The paper deserves publication
Author Response
We thank the Reviewer for appreciating the paper. We are happy that he/she accepted its revised version.

Reviewer 5 Report (New Reviewer)
(1) As a general comment, a global revision of style and clearness of this paper should be done to improve readability. Too much elements are put together and this make the paper hard to read. Furthermore, mathematical details should be put in appendix and only the relevant results should be left in the main sections, i.e., reduce the unnecessary derivation in the text. This would improve the readability of the paper.
(2) Figures and tables should preferably be placed near the first mentioned location. Especially, Fig. 1 is away from its first reference. It is suggested to adjust the structure of figures and tables and their text introduction or explanation.
(3) The selected parameter value of simulation in this study should be specified, whether they are selected by experience or defined artificially. And also, the variables should be represented with different letters or subscripts, like the mathematically described trajectory in subsection 6.2. I think the explanation of parameter selection should be enriched and the representation of variables should be modified.
(4) I was a bit confused regarding Fig. 4. Elements in fig. 4 couldn’t respond to those in formula (20), whether subscript d is the abbreviation of desire, and where are the elements in array E and array responding to fig. 4. Fig. 4 should be revised to improve the intuitiveness of the figure.
(5) I was confused regarding the suitable weights selection for cost function. How to explain the suitable weights selection for properly scaling different terms in the cost functions? That is, how to choose? Why properly scaling different terms? Does different weights selection would lead to different simulation results? I think more details should be included regarding the weights selection.
Author Response
We thank again the Reviewer for the time he/she spent handling our paper. We are glad for the positive comments we received and for all the improvement requests and suggestions he/she provided us. We hope this new version of the manuscript satisfies all the kind comments received.

This manuscript is a resubmission of an earlier submission. The following is a list of the peer review reports and author responses from that submission.
Round 1
Reviewer 1 Report
Overall, the conclusions drawn from the results were logical based on the results. However, the paper suffers significant organizational weak points that require major revision. Several figures and tables were presented prior to initial discussion in the text while others were never discussed in any part of the paper. A significant grammar check is also recommended.
Furthermore, a verification run or any form of reference data for the most optimum combination of controller and optimization algorithm as evidence of the reliability of the model's prediction was not presented, making the paper unacceptable in its current form.
Specific comments:
- In lines 23 to 25, it isn't clear what traditional VTOL robots are. Does this statement compare traditional helicopters (single main rotor and single tail rotor) with quad-rotor drones? I suggest changing the terms "freedom" to "autonomy", "risk" to "need", and "interaction" to "control" as the terms are very vague. "freedom" could be degrees of freedom. "risk" is also very negative to describe "operator interaction". "interaction" is very broad.
- In lines 28 to 30, examples should be referenced.
- Typo error in line 32: Should be Figure 1.
- lines 34 to 35: Again, these applications should have references citing studies that used VTOLs.
- Table 1 should be mentioned first in the text before presentation.
- In Figure 2, delta was not defined.
- In Section 2.2.3, the MSE or accuracy of the model should be presented.
- In line 152, typo error. Should be Equation 15
- Figure 3 was presented before the discussion
- Typo error in Line 165. I suggest use the actual name of the approach such as "PID" instead of A, B, and C
- Figure 5 and Table 4 were not discussed
- Table 3 was presented in the methods section but was not discussed. I suggest placing tables 3 and 4 in the discussion section
- Figure 6 axes and legend labels were too small and not consistent. Use the same range for both x and y axes to easily compare the results among the sub-figures. Furthermore, the legends for the series in figures 6a and 6b are inconsistent; i.e., add PID since the rest have the approaches (NSMC and NBSC) in the legend.
- Section 6.3 are methods
- Section 6.3.2 lacks sufficient discussion.
- The conclusion was in line with the objectives of the paper in comparing controller models and optimization techniques. However, it lacks strength due to the lack of statistical parameters such as MSE or any goodness of fit to determine the model's reliability.
Reviewer 2 Report
This paper tracking autopilots for multirotor aerial robotic system using heuristic nonlinear controllers. Algorithms are proposed and numertical simulations are provided. However, I don't think this paper acceptable to publish in this journal. In the following details my comments:
1. The title indicates that a "multirotor" drone is studied. However, in the content only a quadrotor is studied. The dynamics and control algorithm may differ a little bit between different numbers of rotors. I don't think dynamics and control of a quadrotor can be extended to a multirotor drone.
2. Section 2.2, which discusses the dynamics of the quadrotor, can be shortened, since this topic has been discussed for a long while. Authors can even cite the equations of motion from references without deriving it.
3. Discussion of control laws can be shortened too.
4. Authors mentioned that PID control can be applied to the linearized model of the drone. However, authors didn't explicitly mention the nominal conditions/trajectory which the dynamics model is linearized about. The linearized equations of motion about a hovering state, a cruise state, or a takeoff state could be different, leading to different control strategy or parameters.
5. Although GA and MPSO are employed to tune the control parameters, authors didn't mention how they tune parameters for GA and MPSO. It is known that different fitness function and parameters may lead to different performance of GA and MPSO. Hence, lack of these information could make the simulation results inconclusive.
6. There are several indices to evaluate the performace of tracking response. For example, rise time, steady-state error, overshoot, and so on. Authors didn't employ objective indices for the comparison of control laws in the paper. Hence, I don't see how they may discriminate which control law is better.
Reviewer 3 Report
The paper is accurately written and clear. The results are supported by the series of experiments and visual representation by pictures and graphics.
I have the following major remark to the paper.
The bibliography should be expanded. Why the authors didn't include the sources dated by 2021-2022?
Also please correct: Not all years in the bibliography are bolded. Please keep up the unified style.